# Exometabolomic exploration of culturable airborne microorganisms from an urban atmosphere

Rui Jin[1], Wei Hu[1, 2*], Peimin Duan[1], Ming Sheng[1], Dandan Liu[1], Ziye Huang[1], Mutong Niu[1], Libin Wu[1], Junjun Deng[1], Pingqing Fu[1, 3*]

[1]Institute of Surface-Earth System Science, School of Earth System Science, Tianjin University, Tianjin 300072, China
[2]Tianjin Bohai Rim Coastal Earth Critical Zone National Observation and Research Station, Tianjin University, Tianjin 300072, China
[3]Tianjin Key Laboratory of Earth Critical Zone Science and Sustainable Development in Bohai Rim, Tianjin University, Tianjin 300072, China

*Correspondence to*: Wei Hu (huwei@tju.edu.cn); Pingqing Fu (fupingqing@tju.edu.cn).

**Abstract.** The interactions of metabolically active atmospheric microorganisms with cloud organic matter can alter the atmospheric carbon cycle. Upon deposition, atmospheric microorganisms can influence microbial communities in surface Earth systems. However, the metabolic activities of cultivable atmospheric microorganisms in settled habitats remain less understood. Here, we cultured typical bacterial and fungal species isolated from the urban atmosphere using tryptic soy broth (TSB) and Sabouraud dextrose broth (SDB), respectively, and investigated their exometabolites to elucidate their potential roles in biogeochemical cycles. Molecular compositions of exometabolites were analyzed using ultra-high resolution Fourier transform ion cyclotron resonance mass spectrometry. Annotation through the Kyoto Encyclopedia of Genes and Genomes database helped identify metabolic processes. Results showed that bacterial and fungal strains produced exometabolites with lower H/C and higher O/C ratios compared to both consumed and resistant compounds. As CHON compounds are abundant in both TSB (85%) and SDB (78%), CHON compounds also constituted over 50% of the identified exometabolite formulas. Bacterial strains produced more abundant CHONS compounds (25.2%), while fungal exometabolites were rich in CHO compounds (31.7%). These microbial exometabolites predominantly comprised aliphatic/peptide-like and carboxyl-rich alicyclic molecule (CRAM)-like compounds. Significant variations in metabolites were observed among different microbial strains. Bacteria exhibited proficiency in amino acid synthesis, while fungi were actively involved in amino acid metabolism, transcription, and expression processes. Lipid metabolism, amino acid metabolism, and carbohydrate metabolism varied widely among bacterial strains, while fungi exhibited notable differences in carbohydrate metabolism and secondary metabolism. This study provides new insights into the transformation and potential oxidative capacity of atmospheric microorganisms concerning organic matter at air-land/water interfaces. These findings are pivotal for assessing the biogeochemical impacts of atmospheric microorganisms in clouds or following their deposition.

## 1 Introduction

Bioaerosols include microorganisms (e.g., viruses, bacteria, fungi, archaea), various propagules (e.g., spores and pollen),
biological debris, and biological metabolites (Després et al., 2012; Zhai et al., 2018). They are important components of
aerosols involved in various ecological and atmospheric processes, such as climate change, hydrological, and biogeochemical
cycles (Xie et al., 2021b; Kanakidou et al., 2018; Morris et al., 2011; Fröhlich-Nowoisky et al., 2016; Smets et al., 2016).
Atmospheric microorganisms are usually freely floating or attached to the surface of particulate matter and can be transported
over long distances by wind (Hu et al., 2020; Ruiz-Gil et al., 2020). Bacteria are emitted into the atmosphere from Earth's
surfaces through aerosolization. Due to the harsh environmental stress in the atmosphere, some bacteria develop stress
resistance to survive and degrade low molecular weight organic matter (Šantl-Temkiv et al., 2022; Joly et al., 2015; Ariya and
Amyot, 2004). This resilience gives atmospheric microorganisms a potential role in atmospheric chemistry and physics, driving
chemical reactions at environmental interfaces (e.g., the land-air interface).

Atmospheric bacteria and fungi can maintain metabolic activities due to specific growth characteristics, such as spore
production capacity, ultraviolet resistance, drought resistance, or through extracellular secretions (Huang et al., 2024; Matulová
et al., 2014; Bryan et al., 2019). For example, *Aspergillus niger* resists ultraviolet damage with pigmented spores, and
*Sphingomonas aerolata* NW12 maintains cellular functions by increasing rRNA content and producing proteins while aloft
(Cortesao et al., 2020; Krumins et al., 2014). Airborne microorganisms (especially those in Asian dust) form aggregates with
organic matter, which may serve as a nutrient source, promoting microbial survival and facilitating long distance transport
(Tang et al., 2018; Huang et al., 2024). Atmospheric microbial activity is affected by freeze-thaw and condensation/evaporation
cycles, while microbially produced extracellular polymeric substances (composed primarily of polysaccharides, proteins, and
nucleic acids) help protect cells from osmotic stress (Matulová et al., 2014; Joly et al., 2015). Additionally, a large number of
active microorganisms in air and cloud water are involved in atmospheric chemical processes.

Atmospheric microorganisms play a dual role in atmospheric chemistry and biogeochemical cycles. Not only do they directly
metabolize organic carbon, but they can also reduce the availability of radical sources, thereby decreasing the oxidative
capacity of cloud systems (Zhang et al., 2019; Lallement et al., 2018a; Vaitilingom et al., 2013). Early studies revealed that
some bacteria isolated from cloud water could degrade small organic molecules (e.g., phenol, formaldehyde, carboxylic acids,
and methanol), thereby influencing the organic carbon budget and oxidative balance in clouds (Vaïtilingom et al., 2010; Amato
et al., 2007; Vaïtilingom et al., 2011; Lallement et al., 2018b; Jaber et al., 2020). For example, *Rhodococcus enclensis*, a highly
active species isolated from clouds, biodegrades catechol significantly faster than phenol (Jaber et al., 2020).
Metatranscriptomic analysis and laboratory incubation experiments showed that 93% of 145 bacterial strains isolated from
clouds over Puy de Dôme could degrade phenol (Lallement et al., 2018b). Furthermore, laboratory experiments indicate that

environmental factors such as radiation and pH can influence the biodegradation of organic acids by atmospheric bacteria (Liu et al., 2023b).

Atmospheric microorganisms deposited through wet and dry deposition are a crucial biological source in ocean and freshwater biogeochemical cycles, particularly in oligotrophic systems. Numerous studies highlight the significant role of atmospheric deposition in influencing the nitrogen cycle in marine environments. Mesocosm experiments have demonstrated that the addition of Saharan dust led to a rapid (within 30 hours), pronounced (2–4 fold), and sustained (over 6 days) enhancement in $N_2$ fixation rates, which contributed 3–8% to primary productivity (Rahav et al., 2016b). Aerosol deposition, especially dust,

can deliver active bacteria to the ocean surface and support viable microorganisms like diazotrophs, enhancing $N_2$ fixation and altering marine bacterial communities (Rahav et al., 2016a; Rahav et al., 2018; Na et al., 2023). When airborne microorganisms settle in high-nutrient waters, the dormant spores of autotrophs germinate under favorable conditions and rapidly increase (Genitsaris et al., 2021). These results reveal the potential biogeochemical impacts of the atmospheric deposition of microbes carried by aerosols in surface water.

Wet deposition (e.g., rainfall) brings a wealth of nutrients and microorganisms to surface waters, increasing chlorophyll *a* concentration and enhancing carbon and nitrogen fixation. Short-term wet deposition events play an important role in the temporal variation of new nitrogen production and phytoplankton dynamics in the southeastern Mediterranean, as demonstrated by both experimental and in situ observations (Rahav et al., 2021). Additionally, one notable study demonstrated that rainwater influenced by Saharan dust significantly increased bacterial populations in high-altitude lake water, with rare

taxa becoming dominant based on cultivation experiments (Peter et al., 2014). Despite these findings, there is a lack of research on how atmospheric microorganisms metabolize and degrade organic matter, making it challenging to accurately assess their metabolic processes and potential impacts on atmospheric chemistry and biogeochemical cycles. Therefore, further research is needed to explore the metabolome of atmospheric microorganisms to understand better the essential processes that influence the Earth's atmosphere and ecosystems.

Metabolomics provides direct insights into the function within an ecosystem (Bauermeister et al., 2021). For example, marine phytoplankton generate large amounts of exometabolites, forming a carbon pool known as extracellular release or dissolved primary production (Moran et al., 2022). Microbial exometabolites hold great ecological significance, as they can serve as substrates for other microorganisms, facilitating population growth (Douglas, 2020). For instance, the metabolome of *Pseudomonas graminis* exposed to oxidative stress involves carbohydrate, glutathione, energy, lipid, peptide, and amino acid

metabolism pathways (Wirgot et al., 2019). Furthermore, microbial exometabolites, including extracellular polymeric substances and pigments, are vital for the survival of atmospheric microorganisms, enabling them to withstand long-range transport and extreme conditions in the atmosphere or at the ocean surface (Erkorkmaz et al., 2023; Bryan et al., 2019). Recently, a framework for atmo-ecometabolomics, from sampling to data analysis, was developed to characterize the molecular composition of aerosols (Rivas-Ubach et al., 2019). This framework also provides insights into how aerosol

chemical compositions impact ecosystem structure, function, and biogeochemistry. However, previous studies have not fully explored the role of metabolically active microorganisms in atmospheric aerosols. Deepening our understanding of the

metabolome of atmospheric microorganisms is essential for predicting their ecological roles in biogeochemical cycles and assessing their broader impacts on ecosystem functions and atmospheric processes.

Mass spectrometers with different mass analyzers, including time-of-flight, Orbitrap, and FT-ICR, are favored in many metabolomics applications (Bauermeister et al., 2021). FT-ICR MS is a tool for obtaining high-resolution metabolomic fingerprints and a deeper understanding of potential chemical transformations in various types of samples, such as landfill leachate, algal products, seawater, lake water, soil, rainwater and aerosols (Yuan et al., 2017; Gonsior et al., 2019; Bahureksa et al., 2021; Xie et al., 2021a; Qi et al., 2022). Relevant biosynthetic pathways can be traced through chemical information, relying on the Kyoto Encyclopedia of Genes and Genomes (KEGG) database (Lopez-Ibañez et al., 2021). KEGG is a comprehensive database resource designed to facilitate interpreting high-level functions and systems by integrating molecular-level data, including cellular, organismal, and ecological processes. This database is particularly valuable for analyzing large-scale molecular datasets derived from genomic sequencing and other high-throughput technologies, enabling the exploration of complex biological pathways and networks (Kanehisa et al., 2017). Biochemical pathways were identified by mapping the molecular composition of metabolites obtained by FT-ICR MS to chemical structures in the KEGG COMPOUND Database using the analytical pipeline developed by Ayala-Ortiz et al. (2023). Previously, organic compounds in the mass range of ~ 80–1000 Da have generally been unidentified and overlooked in aerosol particles, yet they may play crucial roles in the ecosystem functioning, especially at the atmosphere-biosphere interface.

This study performed a non-targeted screening of microbial exometabolomes generated during the growth of atmospheric culturable microorganisms, using FT-ICR MS coupled to negative ion mode electrospray ionization (ESI–). The aim was to reveal the characteristics of metabolites from active atmospheric microbes and to underline the important connection between atmospheric microbes and subsurface environments. The specific objectives of this study were to (i) explore the molecular characteristics of exometabolites produced by typical atmospheric culturable bacterial and fungal strains, (ii) elucidate the central metabolic processes and the differences among different strains, and (iii) unravel the potential metabolic capacities of these typical bacteria and fungi. This work provides data to support the roles of atmospheric microorganisms in affecting atmospheric organic matter and biogeochemical processes after deposition onto oceanic or freshwater surfaces.

## 2 Materials and methods

### 2.1 Samples collection and microbial isolation

The sampling site was located on the roof of Building No.19 on the Weijin Road campus of Tianjin University (39.11°N, 117.16°E, about 21 m above the ground level) in an urban area (Fig. S1a) about 500 m from the main road. Total suspended particulate matter (TSP) samples were collected onto quartz fiber filters (10 in. × 8 in.) at a flow rate of 1.05 m$^3$ min$^{-1}$ from 15:00 to 19:00 on 5th, 8th, and 11th January 2022. All samples were stored at –20°C until subsequent treatment. Before sampling, quartz fiber filters were heated in a muffle furnace at 450°C for 6 h to remove organic matter.

Tryptic soy agar (TSA) and Sabouraud dextrose agar (SDA) media were used for bacterial and fungal isolation, respectively. The compositions of the media are detailed in Table S1. All media were sterilized in an autoclave at 115°C for 20 min, and then solid culture plates were prepared under aseptic conditions. Under sterile conditions, one-eighth of each filter sample was cut and placed in a 50 mL centrifuge tube with 40 mL of 1×phosphate buffer saline (PBS) solution. Each sample underwent three parallel experiments. Microorganisms on the filters were detached using low-power ultrasonication in an ice bath for 5 min and then centrifuged at 250 rpm for 30 min. Then, a 100 µL aliquot of the well-mixed suspension was spread-plated onto the two types of solid media. For each suspension, three independent replicates were prepared to ensure that both media were thoroughly coated and evaluated. The plates were incubated at a constant temperature of 37°C for 48 h for bacteria, and at 28°C for 72 h for fungi, with daily growth monitoring (Wang et al., 2023; Timm et al., 2020). From each plate, all phenotypically distinct colonies were picked up and smeared onto fresh media for isolation. Single colonies were picked and re-streaked from each plate at least three times to isolate individual strains (Timm et al., 2020). Multiple streaking isolations are essential to obtain stable and pure strains by gradually diluting the microbial population and allowing for the isolation of individual colonies (Yan et al., 2021; Dunbar et al., 2015).

Isolated strains were preserved using the cryopreservation method. Bacterial strains were cultured in tryptic soy broth (TSB) at 37°C with shaking at 200 rpm, while fungal strains were cultured at 28°C on SDA medium plates under a stationary condition. When the bacteria reached the logarithmic phase, 750 µL of the culture was mixed thoroughly with an equal volume of 50% glycerol, resulting in a final glycerol concentration of 25%, and then stored at –80°C. For fungal strains, once spores were evenly distributed across the plate, they were eluted with 4 mL of sterile PBS buffer. A 750 µL aliquot of the spore suspension was then mixed with an equal volume of 50% glycerol and stored at –80°C with a final glycerol concentration of 25%.

## 2.2 DNA extraction and microbial identification

Liquid cultures were inoculated with single colonies and grown to the logarithmic phase. DNA was extracted from 2 mL liquid cultures using the Universal Genomic DNA Extraction Kit (Solarbio, Beijing, China). The DNA was then stored at –80°C. The 16S rRNA gene region was amplified using 27F (5'-AGAGTTTGATCCTGGCTCAG-3') and 1492R (5'-GGTTACCTTGTTACGACTT-3') primers. The PCR reaction volume was 25 µL, containing 1 µL genomic DNA, 1 µL forward primer (10 µM), 1 µL reverse primer (10 µM), 13 µL 2×Taq PCR Mix (Vazyme, Nanjing, China), and 9 µL ddH$_2$O. The PCR program was: initial denaturation at 95°C for 5 min; followed by 35 cycles of denaturation at 94°C for 45 s, annealing at 55°C for 45 s, extension at 72°C for 1 min; and a final extension at 72°C for 10 min. The fungal internal transcribed spacer (ITS) regions were amplified by PCR (94°C for 5 min; followed by 35 cycles of 94°C for 1 min, 55°C for 1 min, and 72°C for 2 min; final extension at 72°C for 10 min) using the ITS1 (5'-TCCGTAGGTGAACCTGCGG-3') and ITS4 (5'-TCCTCCGCTTATTGATATGC-3') primers. The PCR reaction system for fungi was the same as that for bacteria.

The PCR products were checked by 1% agarose gel electrophoresis and then purified by Universal DNA Purification Kit (TIANGEN, Beijing, China). The sequences of the purified gene fragments were determined by Sanger sequencing (BGI

Genomics Co., Ltd, Beijing, China). Taxonomic assignments were determined from 16S rRNA gene sequences or ITS sequences using the BLAST program at NCBI (https://blast.ncbi.nlm.nih.gov/Blast.cgi).

### 2.3 Microbial growth conditions and metabolite collection

Six representative bacterial strains and six typical fungal strains were selected from the isolated atmospheric microorganisms for subsequent exometabolome studies. Single colonies of all bacterial strains were picked and inoculated in 50 mL TSB medium and incubated at 37°C, 200 rpm for 12 h to obtain the primary seed solution. The primary seed solution was transferred to a 100 mL flask containing 50 mL of sterilized liquid medium at an inoculum level of 3 mL (6%) and incubated for 12 h at 37°C with shaking to obtain the secondary seed solution. The secondary seed solution was also transferred to 50 mL sterilized liquid medium at an inoculum level of 3 mL (6%) and incubated for 7 days at 37°C, 200 rpm. Cultures were then centrifuged at 4500 rpm for 20 min at 4°C. The metabolic product supernatant of about 40 mL was transferred into a new centrifuge tube and stored at –20°C until analysis of molecular composition with FT-ICR MS.

For all fungi strains, spore suspensions were prepared with sterile PBS buffer by eluting spores from fungal plates incubated at 28°C for 7 days. The spore suspension concentration was calculated using a hemocytometer. Then, about $10^8$ cells were inoculated into 50 mL Sabouraud dextrose broth (SDB) and incubated for 15 days at 200 rpm in a 28°C shaker. The method of obtaining and preserving the metabolic products was identical to that used for bacterial cultures.

### 2.4 FT-ICR MS analysis

The metabolic products (2 mL) filtered by 0.22 μm pore membranes were acidified to pH 2 using high-pressure liquid chromatography (HPLC) grade hydrochloric acid (HCl). Dissolved organic matter (DOM) was extracted using a solid phase extraction (SPE) cartridge (200 mg, Oasis HLB, 6cc, Waters, U.S.) to remove salts (Chen et al., 2022; Han et al., 2022). After extraction, the cartridges were dried by flushing with high-purity $N_2$. Finally, 6 mL of HPLC-grade methanol (Sigma-Aldrich) was used to elute the extracted DOM. A 2 mL aliquot of the eluent was collected and stored at –20°C for further analysis. The eluent was analyzed with a 7.0 T superconducting magnet Bruker Solarix FT-ICR MS (SolariX 2xR, Bruker, Germany) equipped with an electrospray ionization (ESI) source in the negative ion mode. The samples were directly injected into the device at a continuous flow rate of 150 μL h$^{-1}$ with a capillary voltage of 5000 V and an ion accumulation time of 0.028 s. The signal acquisition process included 256 cumulative scans and a 4M transient to obtain a higher signal-to-noise ratio (S/N) to resolve the sample fully. More detailed instrument parameter information is available in previous studies (Su et al., 2021). Since FT-ICR MS determines potent polarity molecules with molecular weight (MW) from 100 Da to 1000 Da, pigments or extracellular enzymes secreted by bacteria or fungi can be detected and identified. The complete experimental procedure is shown in Fig. S1b.

### 2.5 Data processing and statistical analysis

The raw datasets were calibrated using the DataAnalysis (ver. 5.0, Bruker Daltonics). The elemental compositions for all recalibrated peaks with S/N ≥ 4, using a mass allowance of ±1.0 ppm, were assigned using Composer (Sierra Analytics, USA) software. The molecular formulas were screened according to the criteria: $C_{0-50}H_{0-100}O_{0-50}N_{0-10}S_{0-3}P_{0-3}$, 0.3 < H/C < 2.5, O/C < 1.2, N/C < 0.5, S/C < 0.2 (Chen et al., 2021b; Yu et al., 2019). Based on the H/C and O/C ratios of the molecular formulas, all formulas were classified into seven categories: lipid-like (1.5 < H/C ≤ 2.0, 0 ≤ O/C ≤ 0.3), aliphatic/peptide-like (1.5 < H/C ≤

2.2, 0.3 < O/C ≤ 0.67), carbohydrate-like (1.5 < H/C ≤ 2.5, 0.67 < O/C < 1.0), unsaturated hydrocarbons (0.67 < H/C ≤ 1.5, O/C < 0.1), carboxyl-rich alicyclic molecule (CRAM)-like (0.67 < H/C ≤ 1.5, 0.1 ≤ O/C ≤ 0.67), aromatic-like (0.2 ≤ H/C ≤ 0.67, O/C < 0.67), and tannin-like/highly oxygenated compounds (HOC) (0.6 < H/C ≤ 1.5, 0.67 ≤ O/C ≤ 1.0) (Bianco et al., 2018).

Depending on the elemental composition of the compound, the molecular formulas were divided into CHO, CHON, CHOS,

CHONS, and CHONSP compounds. For example, CHO indicates that the organic compounds in this category contain only three elements: carbon, hydrogen, and oxygen, with no other elements. The potential metabolic processes of microorganisms were annotated at a molecular level and analyzed using the MetaboDirect pipeline, and the annotation of metabolic pathways mainly relied on the KEGG database (Ayala-Ortiz et al., 2023; Ogata et al., 1999). The molecular mass-based transformation was analyzed by calculating the differences between a pair of m/z for all peaks in each sample and comparing them to the list

of pre-defined masses of common metabolic reactions (biochemical transformations key). The transformation networks encompass processes related to pigment molecules, which are visualized using Cytoscape version 3.10.2. The KEGG database contains three levels of metabolic pathways: 7 classes of primary pathways, 59 classes of secondary pathways, and 563 classes of tertiary pathways (Kanehisa et al., 2017). The Shannon and Chao1 index were calculated in R using the "vegan" (v2.6–4) packages. All statistical analyses were performed in R Studio for R version 4.2.1 and OriginPro 2023b.

**3 Results**

**3.1 Composition of culturable atmospheric microorganisms**

Twenty-four bacterial and sixteen fungal strains were isolated from the TSP samples (Tables S2 and S3). Fig. 1 illustrates the dominant taxa of culturable bacteria and fungi at the phylum and genus levels. The culturable bacterial community primarily consisted of Pseudomonadota (45.8%), Bacillota (37.5%), and Actinomycetota (16.7%). The dominant genera were *Bacillus*

(37.5%), *Pseudomonas* (25%), and *Streptomyces* (16.7%). The isolated bacterial genera represent major taxa commonly found in the atmosphere (Yan et al., 2021; Lee et al., 2017; Calderon-Ezquerro et al., 2021). Gram-positive bacteria such as *Bacillus* and *Streptomyces* possess cell walls predominantly composed of N-acetylmuramic acid. While, Gram-negative bacteria such as *Pantoea*, *Erwinia*, *Stenotrophomonas*, and *Pseudomonas* have cell walls rich in lipopolysaccharides (LPS)/endotoxin (Ruiz-Gil et al., 2020).

In the culturable fungal community, Ascomycota (87.5%) was overwhelmingly dominant. The most abundant fungal genera were *Aspergillus* (37.5%) and *Penicillium* (25%), collectively accounting for over 50% of the isolated strains (Fig. 1). These

findings are consistent with previous studies, which have also identified Ascomycota as the most abundant fungal phylum across different pollution levels and seasons (Abd Aziz et al., 2018; Drautz-Moses et al., 2022; Cáliz et al., 2018; Pyrri et al., 2020). To investigate the potential influence of microbial secondary metabolites in atmospheric and biogeochemical processes,

we selected representative bacterial and fungal strains for further study (Table S4 and Fig. S2). In the following sections, microbial species names are used instead of strain names for conciseness. Our primary focus was on the exometabolome of these atmospheric microorganisms, aiming to elucidate and analyze the metabolic potential and capacity to influence atmospheric biogeochemistry.

## 3.2 Changes of organic molecules in culture media after microbial incubation

The nutrients in the initial culture media supply the growth and reproduction of microorganisms. Through diverse metabolic processes, microorganisms consume substantial amounts of bioavailable organic matter and produce a variety of secondary metabolites. The organic molecular compositions of the media, before and after microbial incubation, were characterized using FT-ICR MS. Organic molecules were grouped into three categories: (1) those present in the initial medium but absent after incubation, indicating consumed organic matter; (2) those absent in the initial medium but present after incubation,

representing biologically controlled organic matter produced by the microorganisms, define as exometabolites; (3) those present in both the initial medium and after incubation, indicating resistant organic matter (Yuan et al., 2017). Some resistant organic molecules may be partially produced by microorganisms and are not classified as exometabolites, as FT-ICR-MS data are not quantitative and cannot unequivocally confirm changes in the abundance of these compounds (Labrie et al., 2022; Noriega-Ortega et al., 2019). Due to limitations in FT-ICR MS techniques, isomers were not considered in this study.

Molecular changes during the metabolic processes of typical atmospheric bacteria (Fig. 2) and fungi (Fig. 3) are demonstrated using van Krevelen diagrams and stacked bar charts, with the formula numbers provided in Table S5. In the bacterial initial medium (TSB), a total of 3416 formulas were detected, with CHON compounds dominating both in number (85.3%) and intensity (92.7%) (Fig. S3a). Furthermore, lipid-like (13%), aliphatic/peptide-like (34%), and CRAM-like (52%) compounds collectively represent 99% of the total number of formulas (Fig. S3b). The bacterial products had a lower H/C ratio (mean ±

SE = $1.43 \pm 0.05$) and higher O/C ratio ($0.46 \pm 0.01$) (Fig. 2a). They are more oxidized than consumed (H/C = $1.57 \pm 0.01$; O/C = $0.32 \pm 0.01$) and resistant molecules (H/C = $1.46 \pm 0.00$; O/C = $0.39 \pm 0.00$). These bacteria consumed 413–1198 formulas, primarily in the categories of lipid-like (19–37%), aliphatic/peptide-like (27–43%), and CRAM-like (31–44%) compounds (Fig. 2b). *Stenotrophomonas* sp. and *Pseudomonas baetica* consumed more aliphatic/peptide-like and CRAM-like compounds (in terms of formula number) than other strain. CRAM-like compounds dominated (54–60%) resistant organic

matter. All bacterial strains produced organic molecules containing predominantly CRAM-like compounds (51–64%) except for *P. baetica* (38%).

In the fungal initial medium (SDB), 3920 formulas were detected, with CHON compounds dominating both in number (77.9%) and intensity (89.4%) (Fig. S3c). Furthermore, lipid-like (14%), aliphatic/peptide-like (32%), and CRAM-like (51%) compounds collectively represent 97% of all molecules in terms of number (Fig. S3d). The formula elemental composition

and category of fungal initial medium is similar to that of bacterial initial medium. The molecules consumed by the fungi were mostly compounds with low O/C ratios (0.34 ± 0.01) and high H/C ratios (1.51 ± 0.02) (Fig. 3a). The fungal products had elevated O/C ratio (0.44 ± 0.02) and reduced H/C ratio (1.32 ± 0.06), which also suggests that airborne microorganisms have the potential to influence atmospheric oxidative capacity and organic carbon budget. The six typical fungi consumed 811–2531 formulas during incubation, with key categories including lipid-like (16–34%), aliphatic/peptide-like (25–32%), and CRAM-like (32–53%) compounds (Fig. 3b). Unlike bacteria, fungi almost exclusively consumed unsaturated hydrocarbons for their growth, with no unsaturated hydrocarbons remaining in the resistant molecules or products. Notably, *Talaromyces* sp. consumed 2531 formulas, more than any other fungal strains, indicating a higher diversity of consumed compounds compared to resistant (1389 formulas) and produced (246 formulas) compounds (Table S5). The resistant compounds for fungal strains were rich in aliphatic/peptide-like (31–37%) and CRAM-like (50–61%) compounds. The product composition of *A. niger* was significantly different, with 20% lipid-like molecules, much larger than the other fungi.

## 3.3 Molecular characteristics of exometabolites from typical cultivable bacteria and fungi

### 3.3.1 Molecular diversity of bacterial and fungal exometabolites

As mentioned above, 651–2868 formulas were produced by the bacterial strains, and 246–1501 formulas were produced by the fungal strains (Table S5), providing a basis for exometabolomic analysis. The molecular diversity of bacterial and fungal exometabolites is illustrated in Fig. S4. For bacteria, the molecular diversity of exometabolites varied considerably among species. The molecular Shannon diversity index of Gram-positive bacteria was 5.64, slightly lower than that of Gram-negative bacteria (6.19). Among the six bacterial strains, *Pantoea vagans*, *Streptomyces pratensis*, and *Stenotrophomonas* sp. showed higher molecular diversity in their exometabolites (Fig. S4a). For fungi, *Rhodotorula mucilaginosa*, *Aspergillus niger*, and *Aspergillus* sp. demonstrated higher molecular diversity of exometabolites (Fig. S4b). The two *Penicillium* species had similar molecular diversity, but the Chao1 index varied considerably. These results indicate the potential metabolic diversity of atmospheric bacteria and fungi.

### 3.3.2 Molecular composition of exometabolites from typical bacterial strains

There are significant differences in the molecular characteristics of exometabolites among the bacterial strains (Fig. 4). The spectral peaks of the exometabolites displayed an uneven distribution across different bacterial strains (Fig. 4a). Overall, CHON compounds accounted for over 50% of the total formula numbers, followed by CHONS compounds, which had an average proportion of 25.2% (Fig. 4a). Given the high abundance of CHON compounds in the initial culture media, CHON compounds were also the primary constituents of the high abundance categories of bacterial exometabolites (Fig. S5a). Moreover, a large number of fractions (18.1–40.6%) of CHONS compounds were produced by the bacterial strains (Fig. 4a and Table S7), much higher than in the initial medium (4.2%). The bacterial exometabolites predominantly consisted of lipid-like, aliphatic/peptide-like, CRAM-like, and tannin-like/highly oxygenated compounds (Fig. 4b).

Significant variations were observed in the exometabolomic profiles of different bacterial species. Notably, *P. vagans* produced a higher proportion of nitrogenous unsaturated molecules (e.g., $C_{12}H_{10}N_2O$, $C_{15}H_{12}N_2O$, $C_{13}H_{12}N_2O$). *Stenotrophomonas* sp. and *P. baetica*, both Gram-negative bacteria, metabolized and synthesized large quantities of CHONS compounds (55.3−61.8% of intensity), largely differed from the other four bacteria strains (only 3.5−21.5% of intensity). Notably, in exometabolites from *Stenotrophomonas* sp., CHONS compounds dominated the lipid-like and aliphatic/peptide-like molecules, distinguishing it from the other bacterial strains (Fig. 4b). Microbes can convert bioavailable lipid-like and protein-like nitrogenous compounds into more oxygenated, unsaturated (more refractory) nitrogenous CRAM/lignin-like compounds (Osborne et al., 2013). Additionally, *Stenotrophomonas* sp. produced a higher number (903 formulas) of high molecular weight compounds (*m/z* >500) (Table S6), likely representing extracellular polymeric substances, e.g., polysaccharides, extracellular enzymes, and cellular debris (Vandana et al., 2023; Moradali and Rehm, 2020).

Comparing the unique molecules in the exometabolites of different strains provides insights into their distinct metabolic processes, and helps to understand the impacts of atmospheric microbial deposition on microbial metabolic activity and the biogeochemical cycle in aquatic ecosystems. A total of 253 common formulas existed among the six typical bacteria, predominantly CHON compounds concentrated in aliphatic/peptide-like and CRAM-like compounds (Fig. 5a and S5b). The unique formulas of each bacterial species contained CHON compounds, with proportions exceeding 25% (Fig. 5a). Most CHON metabolites had an O/N ratio $\leq 3$, suggesting they may be associated with amino and amide groups or N-heterocyclic-containing combinations (Fig. 4c). The metabolism of *P. vagans* appeared to yield unsaturated hydrocarbons compounds containing amino or amide groups, based on molecular classification. *Stenotrophomonas* sp. had the highest percentage (36%) of unique formulas, indicating more diverse metabolic processes (Fig. 5a). Certain *Stenotrophomonas* species can produce polyamines, indole-3-acetic acid ($C_{10}H_9NO_2$), and cytokinins (e.g., $C_{10}H_9N_5O$, $C_9H_8N_4OS$, $C_{17}H_{19}N_5O_5$), which contribute to their ability to protect plants (Peleg and Abbott, 2015; Zhao et al., 2024).

For Gram-positive bacteria, the elemental compositions of the unique formulas were similar, with CHON compounds accounting for more than 50% of all compounds (Fig. 5a). In contrast, for Gram-negative bacteria, CHON compounds were predominant (48.1%) in the unique formulas specific to *P. vagans,* and CHONS compounds are predominantly those specific to *Stenotrophomonas* sp. and *P. baetica*, with proportions of 70.2% and 71.4%, respectively (Fig. 5a). In addition, CHO compounds comprised over 25% of the unique formulas found exclusively in *P. vagans* (Fig. 5a). Notably, among these unique CHO compounds, the most abundant formulas belonged to the $O_9$ class, dominated by $C_{18}H_{28}O_9$ and $C_{13}H_{16}O_9$, indicating higher oxidation states (Fig. S5c). The oxygen numbers of unique CHONS compounds from *Stenotrophomonas* sp. were mainly concentrated in the range of $O_7$ to $O_{11}$ (Fig. S5d), suggesting a higher level of oxidation, probably due to the presence of $-OSO_3H$, $-ONO_2$, or other oxygen-containing groups such as hydroxyl or carboxyl groups. All CHONS compounds specific to *P. baetica* were $CHO_nNS$ (Fig. S5e), with most being $CHO_{10-13}NS$ compounds. These results suggest that these highly oxidized compounds could act as indicator molecules for differentiating among different bacterial species.

### 3.3.3 Molecular composition of exometabolites from typical fungal strains

For fungal exometabolites, CHON compounds also dominated, comprising over 50% of the total formula number on average, followed by CHO compounds (31.7%) (Fig. 6a). The number of formulas identified in fungal exometabolites was lower compared to bacteria (Table S6). Unlike the bacterial strains, which all produced high fractions of CHONS compounds, CHONS compounds appeared only in exometabolites of *R. mucilaginosa*, a yeast species, accounting for approximately 19% of the total formula number (Fig. 6a). This indicates a diversity of sulfur-related metabolic processes for *Rhodotorula*. In contrast, the exometabolites produced by mold fungi, specifically from the genera *Penicillium* (*Penicillium oxalicum* and *Penicillium aurantiogriseum*) and *Talaromyces* (*Talaromyces* sp.), were predominantly CHO compounds, accounting for over 50% of the total exometabolite intensity (Fig. 6a). Among these CHO compounds produced by *Penicillium* and *Talaromyces*, the formulas with the highest intensity were $C_{12}H_{16}O_6$, $C_{15}H_{10}O_6$, and $C_{15}H_8O_7$. The exometabolites from *Talaromyces* sp. were even more specific, with CHO compounds representing 98% of the total intensity and 74% of the formula number.

Fungal exometabolites were mainly composed of aliphatic/peptide-like, and CRAM-like compounds (Fig. 6b). *Talaromyces* sp. produced aliphatic/peptide-like molecules consisting exclusively of CHON compounds, and CRAM-like compounds containing mainly CHO compounds (Fig. S6a). The highest-intensity compounds of *Talaromyces* sp., $C_{15}H_8O_7$ and $C_{16}H_{16}O_5$, belong to aromatic-like and CRAM-like compounds, respectively. *Talaromyces* can produce many bioactive secondary metabolites, with $C_{15}H_8O_7$ and $C_{15}H_{10}O_6$ identified as emodic acids and catenarin, respectively (Lei et al., 2022). These were identified in this study as the leading exometabolites of *Talaromyces* sp, and they are also important products of *Talaromyces avellaneus* and *Talaromyces stipitatus* (Zhai et al., 2016).

Among the exometabolites of the six selected fungi, only 12 formulas were common, including eight CHO compounds and four CHON compounds (Fig. 5b). This is considerably fewer compared to the common formulas observed in bacteria, highlighting the tremendous difference in fungal exometabolites. There were also significant variations in the unique molecule compositions of exometabolites from different fungi. As mentioned above, *R. mucilaginosa* exhibited distinctive molecules characterized by a substantial presence of CHONS compounds (Fig. 5b). A relatively high percentage of these CHONS compounds possess an oxygen number greater than five (Fig. S6b), indicating a high degree of oxidation, with possible functional groups such as amino, amide, and sulfuryl groups (Liu et al., 2021a; Song et al., 2018). Most of the unique formulas in the exometabolites of the six typical fungi were CHON compounds with an O/N ratio ≤ 6, mainly in aliphatic/peptide-like and CRAM-like classes (Fig. 6c), with the possible presence of amino, amide, and nitro function groups. Significantly different from the bacteria, the genera of *Penicillium* and *Talaromyces* generated very distinctive pigmentation during cultivation, with some of these pigment molecules classified as CRAM-like compounds (Fig. S7). Compared to the molecular formulas of pigments resolved in previous studies (Morales-Oyervides et al., 2020; Contreras-Machuca et al., 2022), 10 and 8 typical pigment molecules were identified in the exometabolites from *Penicillium* and *Talaromyces*, respectively (Table S8).

### 3.4 Metabolic processes of typical isolated bacterial and fungal strains

### 3.4.1 Biochemical transformation of bacterial and fungal strains

Microorganisms can selectively uptake substrates and undergo diverse metabolic transformations. During bacterial and fungal growth, there were 168924 and 70646 potential molecular transformations, respectively, with 139798 (83%) transformations unique to bacteria and 41520 (59%) to fungi (Fig. S8a). Among the bacterial transformations, *Stenotrophomonas* sp. had 51667 unique transformations, which accounted for 58% of the total for this strain, demonstrating a higher diversity of metabolite interconversions (Fig. S8b). *P. aurantiogriseum* showed the highest number (24702) of unique transformations, accounting for 65% of its total number of transformations. In contrast, *P. oxalicum* had fewer unique transformations, with 3867 (32%) transformations (Fig. S8c). These results suggest that the potential metabolic pathways vary greatly among *Penicillium* species, highlighting the importance of further studies to characterize the metabolism across different *Penicillium* species.

According to the classification rules of Ayala-Ortiz et al. (2023), 95 and 99 transformation types were identified for bacterial and fungal strains, respectively. Methylation and oxidation/hydroxylation were the most prevalent potential transformations during microbial metabolism (Fig. 7a and 7b). This result is consistent with the metabolic responses of peatland microorganisms to moss leachate and *Pseudoalteromonas* to phage infections (Fudyma et al., 2021; Ayala-Ortiz et al., 2023). Amino acid metabolism differed markedly among bacteria, with alanine, isoleucine, and leucine transformation types being more prominent for *B. subtilis*, while glutamic acid transformation was more abundant for *Stenotrophomonas* sp. (Fig. 7a). For *Talaromyces* sp., carboxylation, glyoxylate, and erythrose transformations were dominant, significantly different from the other fungi (Fig. 7b).

There were significant differences in transformation numbers between bacteria and fungi, with abiotic and amino acid transformations being the most prevalent (Fig. S8d). Notably, cysteine and methionine transformations related to amino acids were significantly higher in bacteria than in fungi (Fig. S8e). In contrast, transformation types associated with phospholipid, biotin, and cytidine were more highly represented in fungi. Therefore, it is hypothesized that bacteria exhibit a stronger inclination towards amino acid synthesis, whereas fungi are more active in transcription and expression.

Transformation networks are frequently used to analyze the metabolic states of organisms since the underlying biochemical reactions are well-known (Plamper et al., 2023). The main organic molecules involved in pigment transformations were CRAM-like and aliphatic/peptide-like compounds for *P. aurantiogriseum* (Fig. 7c) and CRAM-like and aromatic-like compounds for *Talaromyces* sp. (Fig. 7d), respectively. The transformation networks showed interactions within a cluster of CRAM-like metabolites. This suggests that these fungi can efficiently convert certain peptides into stable pigment molecules, which may have industrial applications.

### 3.4.2 Metabolic pathways of typical isolated bacterial and fungal strains

A total of 96 and 53 KEGG metabolic pathways were annotated from typical bacterial and fungal exometabolites, respectively. For bacterial exometabolites, 16 secondary pathways were annotated, with carbohydrate metabolism, xenobiotics biodegradation and metabolism, biosynthesis of other secondary metabolites, and amino acid metabolism being the major pathways (Fig. S9a). For fungi, 12 secondary pathways were annotated, with the major pathways similar to those for bacteria (Fig. S9b), but xenobiotics biodegradation and metabolism were not annotated for fungi.

The primary KEGG Level 3 pathways varied widely among bacteria (Fig. 8a). *P. vagans* and *B. subtilis* were mainly annotated to arachidonic acid metabolism (KO00590) within lipid metabolism. *B. subtilis*, *S. pratensis*, and *Stenotrophomonas* sp. were more oriented towards carbohydrate metabolism, with the main pathways including ascorbate and aldarate metabolism (KO00053), amino sugar and nucleotide sugar metabolism (KO00520), and pentose and glucuronate interconversions (KO00040). For *Stenotrophomonas* sp., amino acid metabolism of tyrosine metabolism (KO00350) and histidine metabolism (KO00340) were more enriched. Bacteria and fungi shared 16 pathways, mainly related to carbohydrate metabolism, energy metabolism, and membrane transport, which are essential metabolic processes that sustain cell growth and reproduction.

The six typical fungal strains can be classified into different groups according to the KEGG Level 3 pathways (Fig. 8b). *R. mucilaginosa* and *Penicillium* species were mainly involved in carbohydrate metabolism, specifically fructose and mannose metabolism (KO00051) and galactose metabolism (KO00052). Additionally, they were characterized by lysine biosynthesis (KO00300), ABC transporters (KO02010), and diterpenoid biosynthesis (KO00904). *Talaromyces* sp. was predominantly involved in aflatoxin biosynthesis (KO00254), while *Aspergillus* metabolism was dominated by arginine and proline metabolism (KO00330), methane metabolism (KO00680) and cyanoamino acid metabolism (KO00460).

Analytical approaches that map molecular composition to microbial metabolic pathways offer a powerful framework for understanding the complex metabolic functions and key transitions in microbial processes. This helps to elucidate the unique metabolic processes by which microorganisms survive and adapt in atmospheric environments. The metabolic processes of bacteria and fungi in the surface Earth system are closely linked, which is overlooked in this study, and further elucidation of these connections will enhance our understanding of the impact of microorganisms on atmospheric environments and biogeochemical processes.

## 4 Discussion

### 4.1 Survival mechanisms of culturable atmospheric microorganisms

At the genus level, culturable atmospheric bacteria isolated from the aerosol samples mainly included *Bacillus*, *Pseudomonas*, *Streptomyces*, and *Pantoea* (Fig. 1). These bacterial taxa were predominant across urban, rural and forest aerosols, displaying strong environmental adaptability (Nunez et al., 2021; Gusareva et al., 2019; Souza et al., 2021). Their widespread distribution suggests remarkable resilience to harsh environmental stressors, enabling them to survive and maintain culturability (Hu et al., 2018; Hu et al., 2020).

Most bacterial genera cultured from the atmosphere in various worldwide investigations are Gram-positive bacteria, with *Bacillus* as one of the most abundant genera (Yoo et al., 2019; Maki et al., 2010; Hua et al., 2007). The *Bacillus* genus has often been identified as a dominant taxon in dust aerosols (Rossi et al., 2024; Péguilhan et al., 2023; Maki et al., 2022). For instance, *B. subtilis* has been found to increase significantly during dust events and maintain culturability. *B. subtilis* is known for its high adaptability to diverse extreme conditions, e.g., salt- and temperature-tolerant, allowing it to survive in a wide range of environments (Losick, 2020; Liu et al., 2018; Maki et al., 2022). The versatile metabolic capacity may play a

physiological role in facilitating bacterial survival under harsh environments. Previous studies demonstrated that *Bacillus* sp. 3B6 was able to efficiently biotransform sugars into extracellular polymeric substances to protect this bacterium under hostile environment conditions, including cold temperature, radical exposure, and freeze processes (Matulová et al., 2014). Additionally, members of the *Bacillus* genus are known for their ability to form endospores, enabling them to withstand atmospheric stressors and survive long-range transport (Nicholson et al., 2000).

*Pantoea* is a highly diverse genus found in both aquatic and terrestrial environments (Walterson et al., 2015; Murillo-Roos et al., 2022). *Pantoea* species exhibit a robust oxidative stress response by synthesizing enzymes such as catalase, superoxide dismutase, and peroxidase (Dahiya et al., 2024; Tambong, 2019). *Pseudomonas* species are well-adapted to atmospheric conditions, and they often oxidize common atmospheric volatile organic compounds, including formaldehyde and methanol (Husarova et al., 2011). *Streptomyces* is a dominant genus in aerosols from various megacities and background areas, especially in winter (Chen et al., 2021a; Petroselli et al., 2021; Li et al., 2019). Furthermore, *Streptomyces* has been widely reported in aerosols from European deserts and urban sewage treatment plants (Núñez et al., 2024; Zhan et al., 2024). *Streptomyces* is a drought-tolerant bacterial genus with spore-forming ability (Taketani et al., 2016). Their spores can enter a stable and quiescent state under environmental stress, enabling survival in diverse natural conditions (Naylor et al., 2017).

In this study, *Aspergillus* and *Penicillium* were the dominant fungal genera isolated. The spores of these species contain pigments, and most fungi produce secondary metabolites that include various pigments. Specifically, the exometabolites of *Penicillium* and *Talaromyces* were enriched with pigment compounds, particularly CRAM-like molecules with CHO elemental compositions (Fig. 6a and 6b). Some aliphatic/peptide-like and aromatic-like compounds were biotransformed into CRAM-like pigment molecules ($C_{14}H_{10}O_6$, $C_{23}H_{24}O_7$, $C_{26}H_{29}NO_8$, $C_{14}H_{10}O_6$, and $C_{15}H_{10}O_5$) (Fig. 7c and 7d). Microbial pigments are chemically diverse, including flavonoids, isoprenoids, porphyrins, N-heterocyclics, and polyketides (Contreras-Machuca et al., 2022; Morales-Oyervides et al., 2020; Venkatachalam et al., 2018). Recent studies have highlighted the potential of *Penicillium* and *Talaromyces* as robust producers of natural pigments, noted for their thermal, pH, and light stability (Ugwu et al., 2021; Akilandeswari and Pradeep, 2016). Many microorganisms isolated from atmospheric aerosols and rainwater were highly pigmented, potentially aiding their survival under intense ultraviolet exposure and photo-oxidative stress (Tong and Lighthart, 1997; Fahlgren et al., 2010; Fahlgren et al., 2015; Dassarma and Dassarma, 2018; Ziegelhoffer and Donohue, 2009). Additionally, many culturable halotolerant ice-nucleating bacterial and fungal strains isolated from coastal precipitation and aerosols could produce pigments (Beall et al., 2021). Microbial pigments can act as a protective layer for the cells, contributing to their long-distance atmospheric transport and adaptation to extreme environments.

**4.2 Potential impacts of atmospheric microbial metabolites on atmospheric and marine organic matter**

Molecules produced by culturable atmospheric microorganisms predominantly consisted of compounds with high O/C ratios and low H/C ratios, indicating a higher degree of oxidation (Fig.2 and 3). These findings align with previous studies, which suggest that organic compounds with lower O/C and higher H/C ratios are preferentially decomposed during microbial biotransformation, resulting in the production of oxygen-rich structures (Che et al., 2021; Gu et al., 2023). As microorganisms

metabolism processes break down complex organic matter, oxygen-containing functional groups accumulate, signifying
oxidative processes (Yuan et al., 2017).

Microorganisms consume molecules with high H/C ratios and labile molecules, leading to the generation of lower molecular weight organic molecules with lower H/C ratios (Ni et al., 2024). The decreased H/C ratios indicate a reduced presence of aliphatic hydrocarbons, suggesting that these microorganisms may contribute to forming complex compounds with lower volatility (Li et al., 2021; Abbasian et al., 2015). These results underscore microorganisms replace H atoms with oxygen-rich
functional groups such as carboxyl (COOH) and aldehyde (COH) groups during the oxidative degradation of DOM (Labrie et al., 2022). Therefore, the elevated O/C ratios indicate that atmospheric microorganisms may significantly enhance the oxidation of organic matter in clouds and aerosols, potentially influencing the organic carbon budget in atmospheric systems. The bacterial exometabolites predominantly consisted of lipid-like, aliphatic/peptide-like, CRAM-like, and tannin-like/highly oxygenated compounds (Fig. 2b). In contrast, the fungal exometabolites were mainly aliphatic/peptide-like and CRAM-like
compounds (Fig. 3b). Notably, CRAM-like compounds were ubiquitous and the most abundant across both bacterial and fungal exometabolites, underscoring their pivotal role in microbial metabolic processes. CRAM-like compounds are an essential component of marine DOM and are produced and consumed by heterotrophic bacteria, which transform recalcitrant DOM (rDOM), as demonstrated in laboratory studies of the model diatom *Skeletonema dohrnii* (Liu et al., 2023c; Liu et al., 2021b). Additionally, long-term (220 days) macrocosm experiments at 16–22°C with mixed cultures of coastal seawater and diatom
lysate have demonstrated that bacteria and archaea are key producers of rDOM, with CRAM-like compounds being a central component (He et al., 2022). Our findings align with these results, confirming that CRAM-like compounds are significant exometabolites of common atmospheric microorganisms. Consistent with these findings, incubation studies conducted on cloud water collected at the Puy de Dôme Observatory at 15°C in the dark revealed that microbial products are predominantly composed of lipid-like (16.5%), aliphatic/peptide-like (14.3%), and CRAM-like (44.7%) compounds, which closely aligns
with the results of this study (Bianco et al., 2019). Moreover, after dark incubation for 30 days at room temperature, rainwater from coastal storms produced molecules primarily in the form of aliphatic/peptide-like and CRAM-like compounds (Mitra et al., 2013), similar to the products of microorganisms in this study. These results suggest that atmospheric microorganisms could contribute to the biotransformation of rDOM as it descends into marine environments.

**4.3 Diverse metabolic processes and products of atmospheric microorganisms**

Atmospheric microorganisms, including various bacterial and fungal strains, exhibit remarkable diversity in their metabolic processes and the products they generate. Analysis of exometabolites from different microbial species has revealed distinct molecular characteristics that underscore their unique metabolic pathways. The metabolomic profiles of airborne bacteria showed notable differences in metabolite production; however, CHON compounds accounted for over 50% of the total molecular formulas identified for all strains (Fig. 4). Similar findings were observed in the actual atmospheric environment,
that the interactions between alive bacteria (*B. subtilis*, *Pseudomonas putida*, and *Enterobacter hormaechei*) and ·OH radicals in clouds contributed a significant amount of CHON compounds, comprising over 50% of the water-soluble compounds (Liu et al., 2023a).

A large number of CHONS compounds, belonging to aliphatic/peptide-like molecules, were detected in the metabolites of the Gram-negative bacteria *Stenotrophomonas* sp. and *P. baetica*, accounting for 40.6% (55.3% of intensity) and 33.2% (61.8% of intensity), respectively (Fig. 4b and S5). These compounds, including extracellular polymeric substances, play a crucial role in microbial survival by forming protective biofilms and contributing to the atmospheric biogeochemical cycle. The most frequently detected genus in the atmosphere, *Pseudomonas*, is a key producer of biosurfactants, such as rhamnolipids, syringafactin, and viscosin (Carolin C et al., 2021; Eras-Muñoz et al., 2022; Liu et al., 2013). Several potential rhamnolipid molecules, such as $C_{12}H_{18}O_7$, $C_{12}H_{20}O_7$, and $C_{16}H_{26}O_7$, were detected in the exometabolites. These biosurfactants can impact atmospheric chemistry (e.g., secondary organic formation) and modify cloud microphysics by enhancing cloud condensation nuclei activation owing to their exceptional ability to reduce surface tension (Delort et al., 2010).

Building on prior studies, some *Stenotrophomonas* spp. are known to synthesize a range of biologically active compounds, including a variety of antibiotic enzymes such as β-lactams, aminoglycosides, and macrolides, alongside chitinases, lipases, and proteases (Ryan et al., 2009; Peleg and Abbott, 2015; Wang et al., 2018). These findings suggest that many of the CHONS compounds identified in the exometabolites are likely proteases or their degradation products, underscoring the pivotal role of *Stenotrophomonas* in protein turnover and nutrient cycles in the land/water-air interfaces (Liu et al., 2016). Furthermore, the molecular composition of the exometabolites pointed to an enrichment of amino acid metabolic pathways, like tyrosine metabolism (KO00350) and histidine metabolism (KO00340) (Fig. 8a). These pathways, coupled with the known reactivity of amino acids, highlight the importance of considering the amino acid presence and transformation when investigating the chemical composition and oxidative processes occurring within cloud water (Bianco et al., 2016). This deeper understanding of bacterial exometabolites in atmospheric conditions provides new insights into the role of microorganisms in cloud chemistry and broader biogeochemical cycles.

Significant variations were observed in the metabolic products of different fungi: yeasts produced the highest diversity of molecules, while molds produced more homogeneous compounds, with certain molecules having very high intensities (Fig.6). *R. mucilaginosa*, as an essential biotechnological yeast, is a typical platform strain that can produce many functional bioproducts (Li et al., 2022). *R. mucilaginosa* was predominantly involved in carbohydrate metabolism (Fig. 8b). In this study, several saturated fatty acids, including myristic acid ($C_{14}H_{28}O_2$), stearic acid ($C_{18}H_{36}O_2$), and palmitic acid ($C_{16}H_{32}O_2$), and unsaturated fatty acids including linoleic acid ($C_{18}H_{32}O_2$) and linolenic acid ($C_{18}H_{30}O_2$), were detected. These organic acids were also abundantly present in atmospheric aerosols and may affect the atmospheric chemical and physical processes, e.g., cloud formation (Mkoma and Kawamura, 2013; Balducci and Cecinato, 2010; Raymond and Pandis, 2002). Notably, palmitic acid, stearic acid, and linoleic acid are potential molecular tracers for estimating the contribution of cooking emissions to organic aerosols (Cheng and Yu, 2020; Ma et al., 2023). However, our findings reveal that fatty acids were also abundant in the metabolites of *Rhodotorula*, suggesting that microorganisms may contribute to atmospheric fatty acid levels. This highlights the need to consider microbial sources when assessing the composition of atmospheric organic aerosols.

Among airborne fungi, the genera *Aspergillus*, *Penicillium*, and *Talaromyces*, which are classified as mold fungi, are acknowledged as some of the most chemically prolific organisms. These genera can synthesize a wide variety of secondary

metabolites (exometabolites) (Frisvad, 2015; Zhai et al., 2016; Morales-Oyervides et al., 2020). They produce specialized compounds such as polyketides, non-ribosomal peptides, and terpenes, further showcasing their metabolic versatility and biochemical complexity (Frisvad, 2015; Adelusi et al., 2022). This remarkable chemodiversity highlights their ecological significance in atmospheric biogeochemistry. *A. niger* is ubiquitous in the environment and plays a significant role in the global carbon cycle (Baker, 2006; Schuster et al., 2002; Šimonovičová et al., 2021). It is rich in genetic and metabolic diversity, acting as one of the most important microorganisms used in biotechnology (Sun et al., 2007; Jørgensen et al., 2011). In its natural growth state, *A. niger* possesses large cryptic biosynthetic gene clusters (BGCs), which synthesize a wide range of extracellular enzymes to degrade special biopolymers in the environment, thus allowing the fungus to obtain nutrients (Yu et al., 2021; Romsdahl and Wang, 2019). Anthraquinones like $C_{15}H_{12}O_5$, $C_{15}H_{10}O_5$, and $C_{15}H_{10}O_6$ were detected in the product of *Penicillium* and *Talaromyces*, and these compounds can protect symbiotic plants from pests or pathogens (Etalo et al., 2018). This metabolic plasticity underscores their ecological importance and the potential for biotechnological applications in various industries. Given the ecological and biotechnological significance of secondary metabolites from fungi, further exploration of these exometabolites remains a priority in atmospheric microbial research.

## 5 Conclusions

This study unveiled the molecular characteristics of exometabolites produced by typical culturable microorganisms in the urban atmosphere, integrated with metabolic pathways, enriching the foundational data for untargeted studies of microbial exometabolites using FT-ICR MS. Vital metabolic processes for various bacteria and fungi were identified through mass-based transformation analysis and KEGG enrichment analyses. The culturable atmospheric bacteria were primarily *Pseudomonas* and *Bacillus*, and the predominant fungal genera were *Aspergillus* and *Penicillium*. Bacterial and fungal strains produce exometabolites with lower H/C ratios and higher O/C ratios, suggesting that airborne microorganisms have the potential to influence atmospheric oxidative capacity and the organic carbon budget. The typical bacteria largely contributed to CHON and CHONS compounds in their metabolism, whereas fungi mainly produced CHO compounds as exometabolites. The microbial exometabolites predominantly consisted of aliphatic/peptide-like and CRAM-like compounds. The unique molecules of each typical bacterial strain were highly oxidizable and can serve as indicator molecules for various bacteria. Micrbial metabolic pathways varied significantly, with lipid metabolism and amino acid metabolism showing considerable variations among different bacteria, while carbohydrate metabolism and secondary metabolism differed markedly among fungi. *Talaromyces* sp. exhibited a unique metabolic process with exometabolites dominated by CHO compounds, primarily undergoing carboxylation and glyoxylate transformations. Additionally, aflatoxin biosynthesis was identified as the central metabolic pathway of *Talaromyces* sp., along with the biosynthesis of other secondary metabolites.

These findings enhance our understanding of the potential roles of atmospheric microorganisms in atmospheric chemistry and biogeochemical cycles, particularly by supporting the occurrence of metabolic processes when atmospheric microorganisms are deposited onto oceanic or freshwater surfaces. Future research should focus more on the deposition and activity of

atmospheric microorganisms in natural environments and develop a comprehensive database of source profiles for typical
atmospheric microbial exometabolites at the molecular level.

*Data availability.* The dataset for this paper is available upon request from the corresponding authors (huwei@tju.edu.cn; fupingqing@tju.edu.cn).

*Author contribution.* WH, PF, and RJ designed the study and the experiments. RJ, MN, and PD executed experiments. RJ, MS,
DL, and ZH analyzed the data. RJ and WH visualized the data, and wrote the draft manuscript with input from all of the authors. All of the authors discuss, edit, and approved the manuscript.

*Competing interests.* The authors declare that they have no conflict of interests.

*Acknowledgments.* We thank Chao Ma and Shujun Zhong for their help in the FT-ICR MS data processing.

*Financial support.* This study was supported by the National Natural Science Foundation of China (Grant Nos. 42221001,
42130513, and 42394151).

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

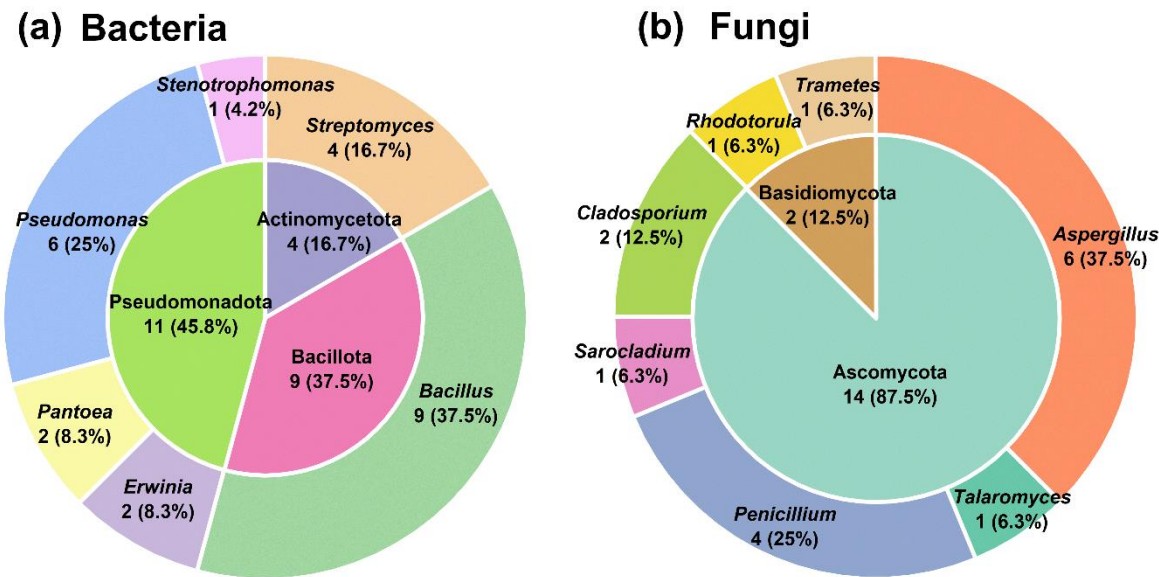

**Figure 1.** The distribution of culturable bacterial (a) and fungal (b) taxa at the phylum (inner pie charts) and genus (external pie charts) levels.

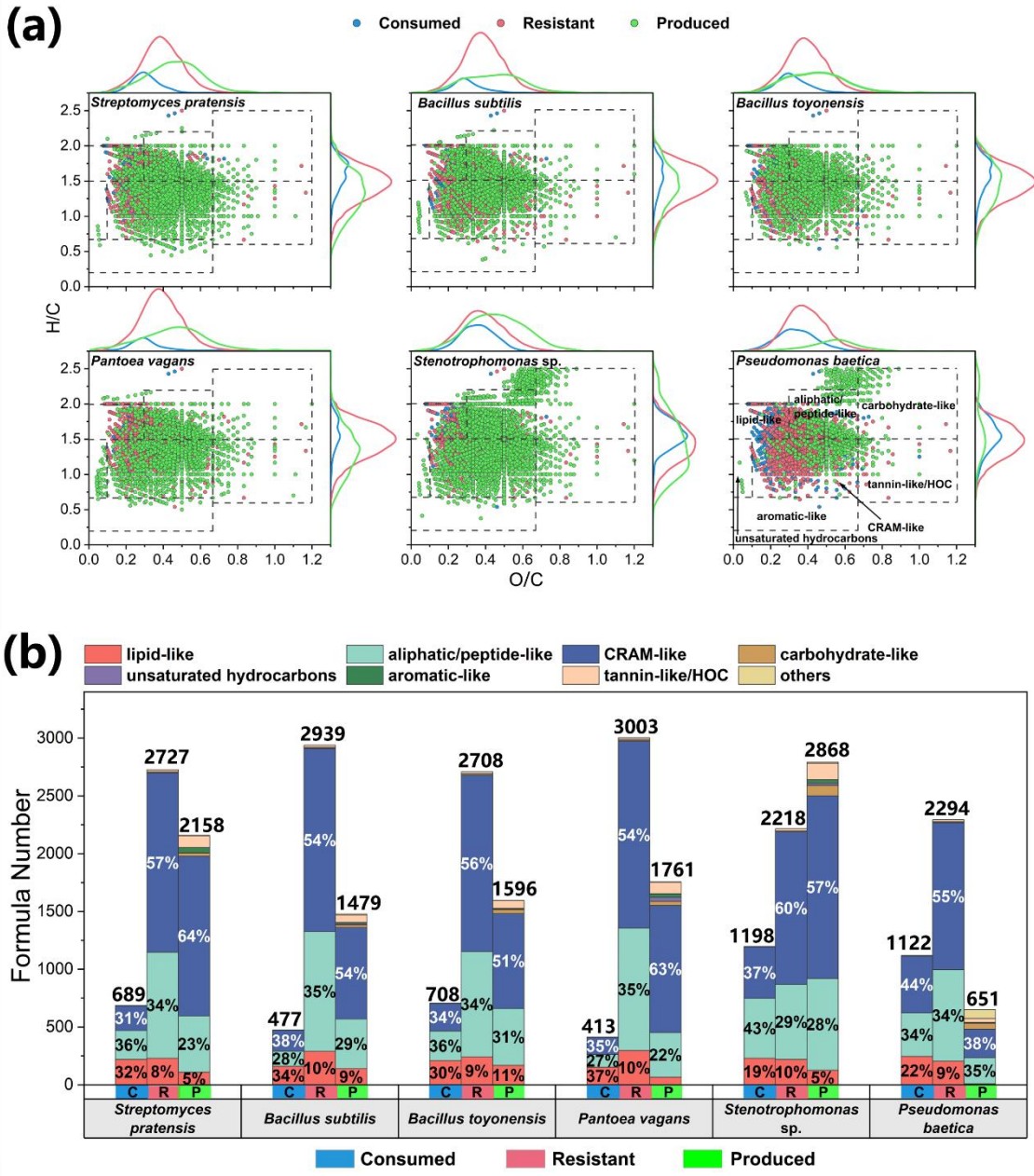

**Figure 2.** Changes in molecular compositions of culture media during the metabolic processes of six typical atmospheric culturable bacteria. (a) Van Krevelen diagrams illustrate changes in different categories of organics in the media after incubation. The top three van Krevelen diagrams represent the chemical composition of Gram-positive bacteria, the bottom three van Krevelen diagrams correspond to Gram-negative bacteria. (b) The stacked bar charts indicate the formula numbers of different categories of consumed (labeled as C), resistant (R), and produced (P) organic matter.

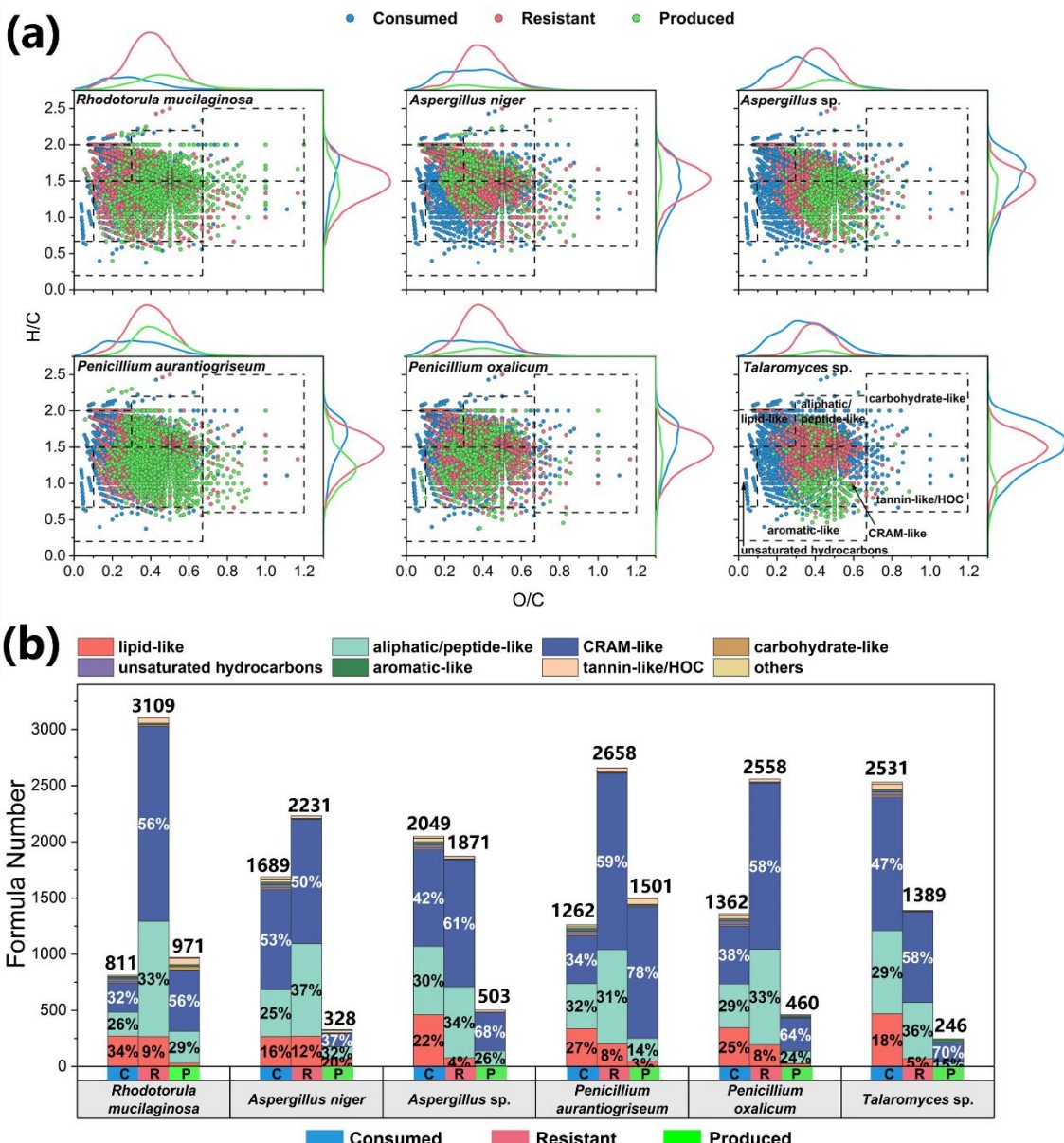

**Figure 3.** Changes in molecular compositions of culture media during the metabolic processes of six typical atmospheric culturable fungi. (a) Van Krevelen diagrams illustrate changes in different categories of organics in the media after incubation. (b) The stacked bar charts indicate the formula numbers of different categories of consumed (labeled as C), resistant (R), and produced (P) organic matter.

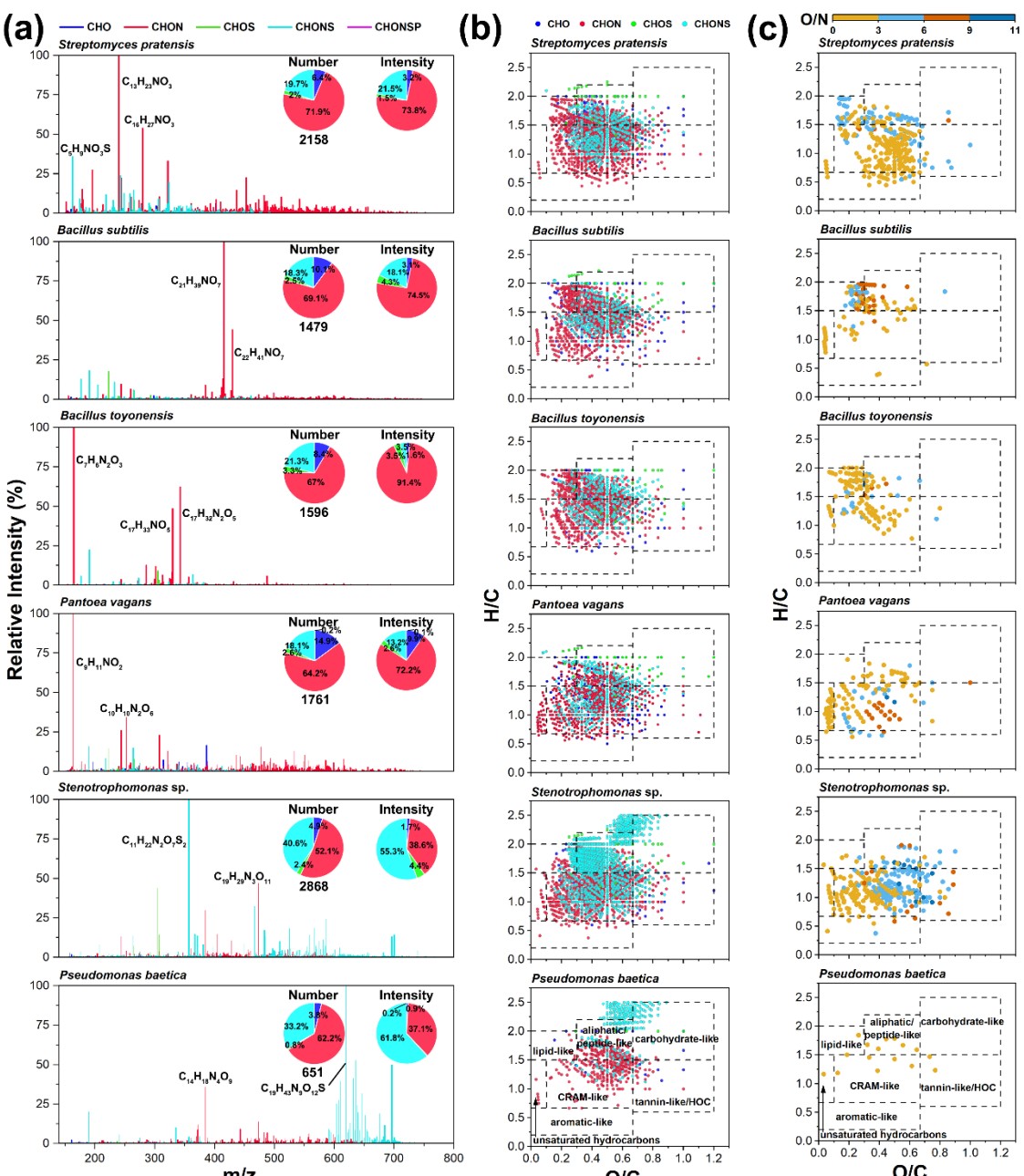

**Figure 4.** Molecular characteristics of bacterial exometabolites. (a) Mass spectra and pie charts show the distribution of m/z values and elemental compositions in bacterial exometabolites. (b) Van Krevelen diagrams show the molecular distribution of bacterial exometabolites classified by elemental composition (CHO, CHON, CHOS, and CHONS). (c) Van Krevelen diagrams reveal the distribution of unique CHON compounds with different O/N ratios for each bacterial strain.

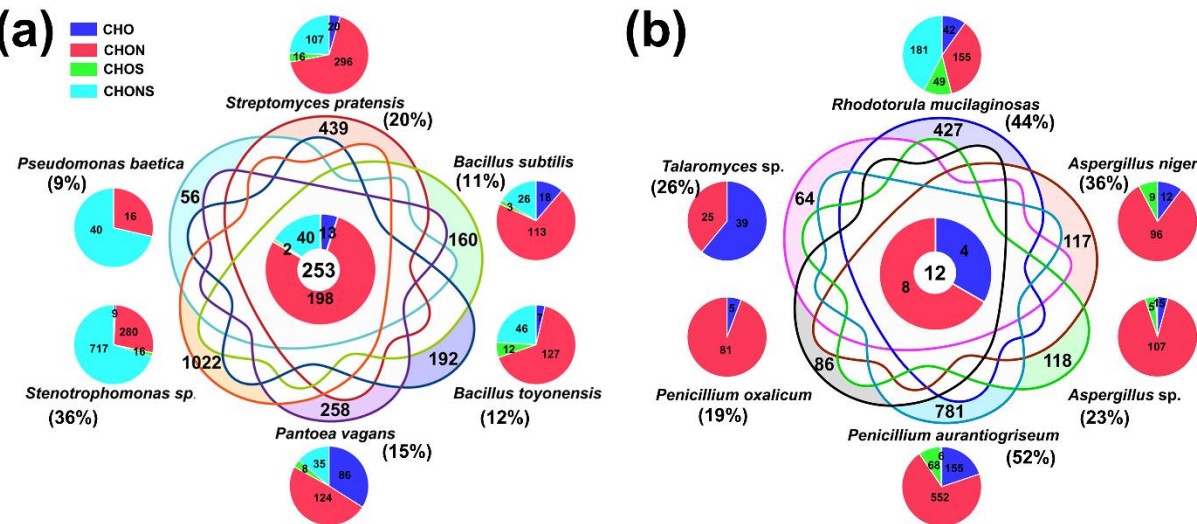

**Figure 5.** Differences in exometabolites among different typical bacteria (a) and fungi (b). The Venn diagrams display the numbers and percentages of shared and unique formulas for six typical bacterial or fungal strains. The percentages represent the number fraction of unique formulas relative to the total formulas for each strain. The donut diagram in the center and pie charts show the elemental compositions of shared and unique formulas for each strain.

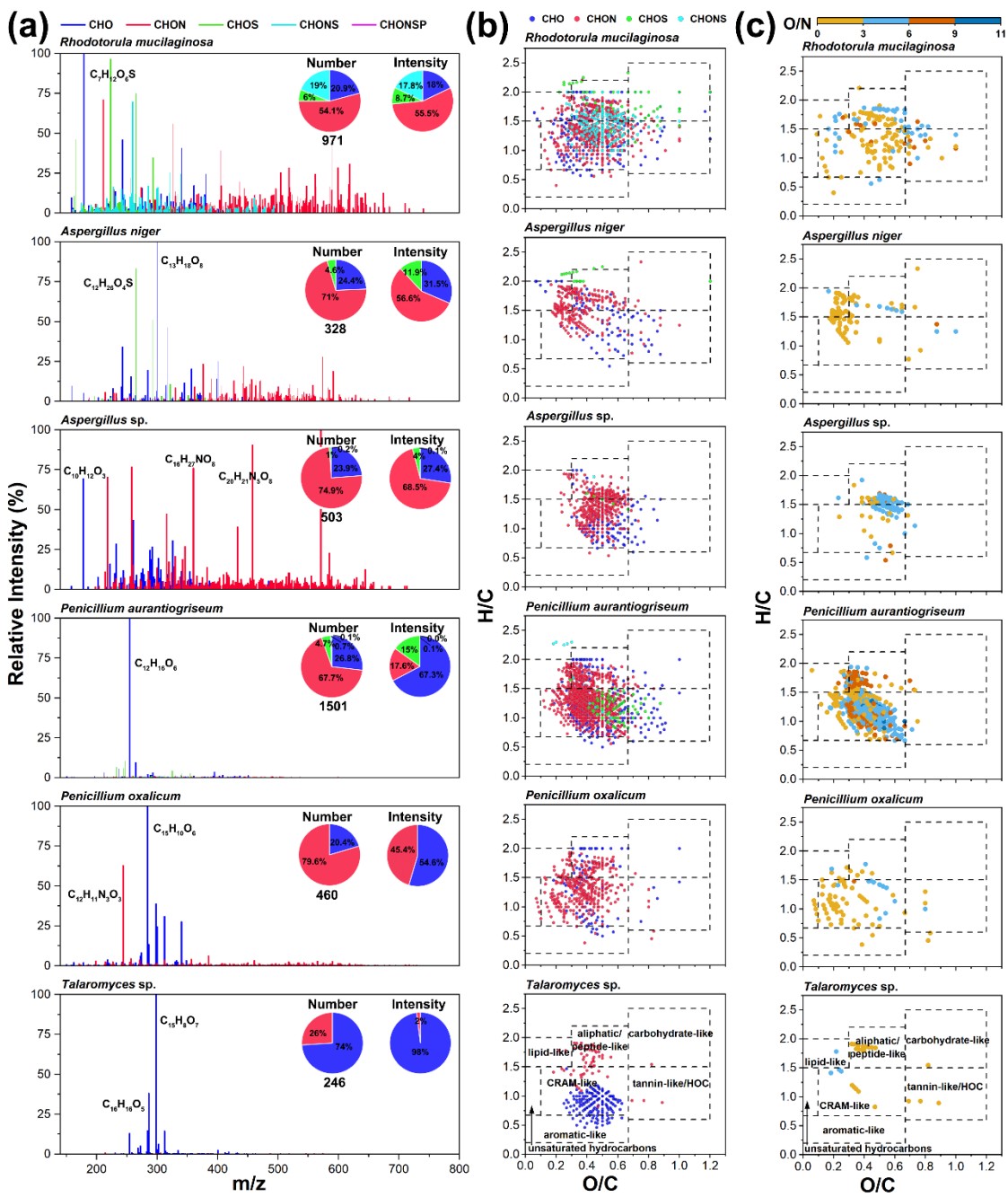

**Figure 6.** Molecular characteristics of fungal exometabolites. (a) Mass spectra and pie charts show the distribution of *m/z* values and elemental compositions of fungal exometabolites. (b) Van Krevelen diagrams show the molecular distribution of fungal exometabolites classified by elemental composition (CHO, CHON, CHOS, and CHONS). (c) Van Krevelen diagrams reveal the distribution of unique CHON compounds with different O/N ratios for each fungal strain.

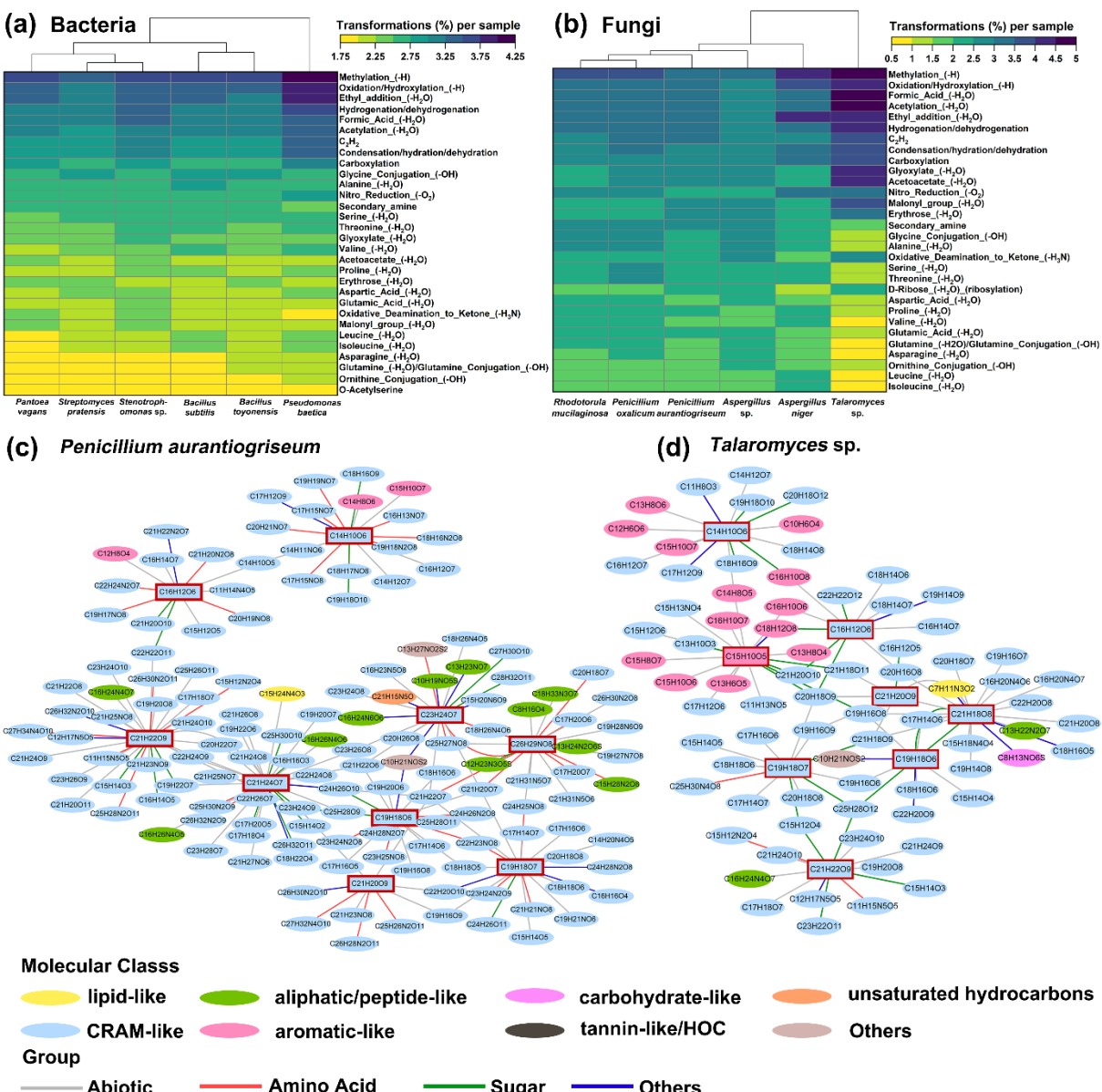

**Figure 7.** The types and networks of biochemical transformations for bacteria and fungi. Heatmaps show the top 30 transformation types in terms of relative abundance for bacteria (a) and fungi (b). Molecular networks illustrate the transformations for the major pigment formulas (Table S8) of *P. aurantiogriseum* (c) and *Talaromyces* sp. (d).

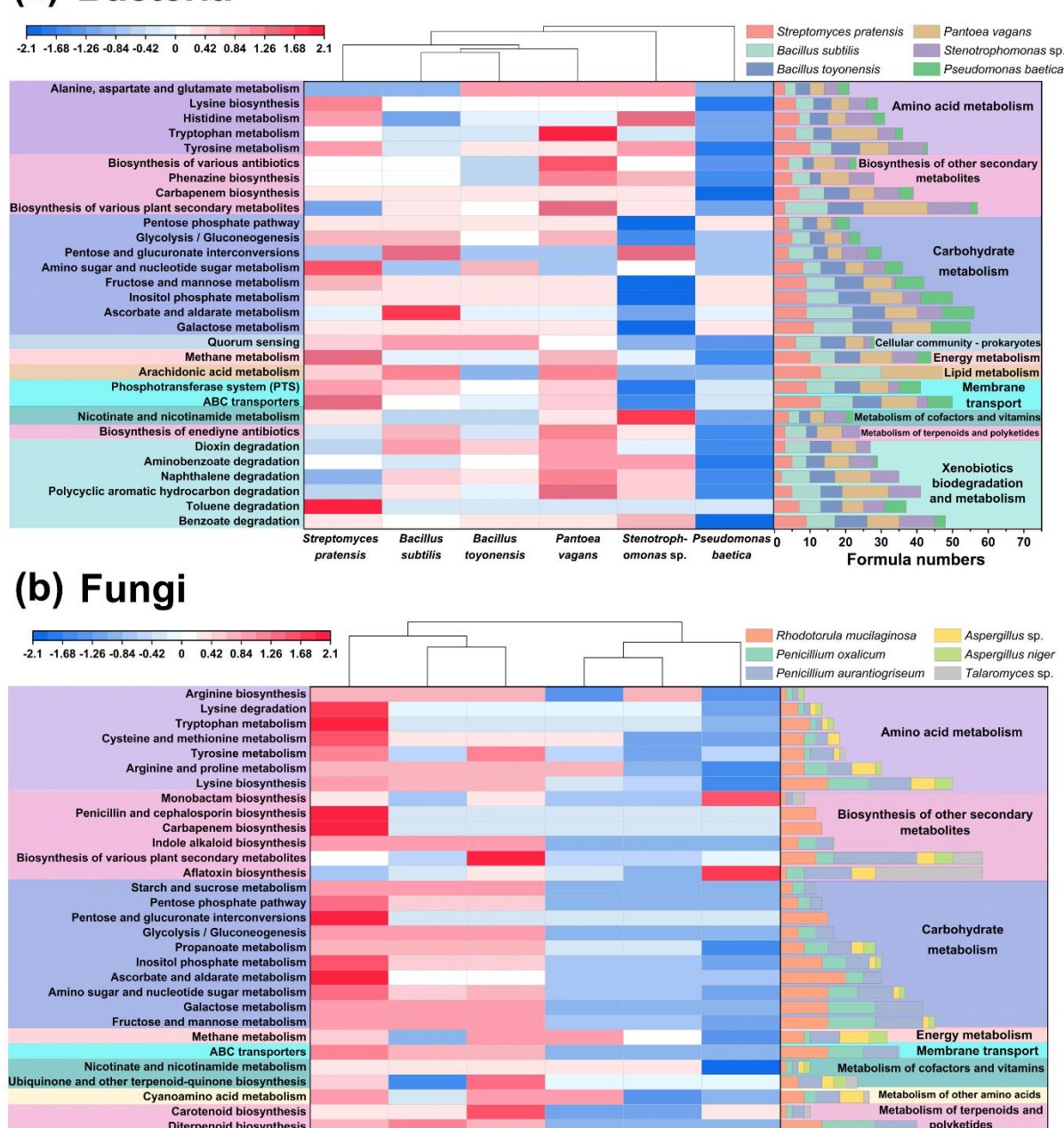

**Figure 8.** The KEGG metabolic pathways for typical airborne bacterial (a) and fungal (b) strains. Heatmaps illustrate the normalized relative abundance (top 30) of KEGG tertiary pathways of bacterial and fungal strains. Stacked bar diagrams show the formula numbers annotated to microbial exometabolites in each pathway.