# Peer review of "Exometabolomic exploration of culturable airborne microorganisms from an urban atmosphere"

_EGUsphere, 2024_

## Author Comment (AC1)

Dear Editor and Reviewers,

We thank the referees for their careful reading and constructive comments, which have allowed us to make further enhancements and significantly raised the quality of the manuscript. Our responses to the comments from the three referees are provided below. The reviewers' comments are shown in blue, while our responses are in black. All page and line numbers mentioned in the response refer to the revised version of the manuscript. Revisions are marked in blue in the revised manuscript. The revised manuscript and supplement (see marked copy) are provided below the response.

Sincerely,

HU Wei (on behalf of co-authors)

**Response to Referee 1:**

This work from Jin et al., titled with "Exometabolomic exploration of culturable airborne microorganisms from an urban atmosphere" has investigated the metabolic activity of airborne culturable microorganisms at the molecule level. Distinct metabolic profiles were observed for the different bacterial strains and the fungal strains. Some suggestions are as below:

R1 Comment 1: It is interesting to reveal the different metabolic profile of microorganisms isolated from the ambient air. How about the abundance of these unique "produced" products in the ambient air?

**Response:** Thank you for your valuable comments. It is feasible that atmospheric microorganisms could contribute to biogeochemical processes through metabolic activities, including carbon fixation or degradation of carbon compounds (e.g., pollutants). In this study, analysis of atmospheric microbial exometabolites revealed many molecules that existed in ambient aerosols, such as myristic acid ($C_{14}H_{28}O_2$), stearic acid ($C_{18}H_{36}O_2$), palmitic acid ($C_{16}H_{32}O_2$), linoleic acid ($C_{18}H_{32}O_2$), linolenic acids ($C_{18}H_{30}O_2$) and other lipid-like molecules.

We agree that the understanding on the abundance of these unique exometabolites produced by airborne microorganisms in ambient air is an important aspect of their ecological role. However, our current study primarily focused on characterizing the metabolic profiles of the isolated microorganisms under controlled conditions to assess their potential impacts on atmospheric chemistry and biogeochemical processes. However, because of technical limitations, the metabolite analysis methods used in the study can perform qualitative and semi-quantitative analyses but cannot provide accurate quantitative results. In future metabolomics analyses of ambient air samples, we need to explore the abundance of microbial exometabolites by using high-resolution quantitative analytical methods. Our current findings can serve as a database of atmospheric microbial exometabolites to better assess their ecological significance in the atmosphere.

R1 Comment 2: The introduction is somewhat lengthy; it is recommended to condense and refine it appropriately.

**Response:** Thank you for your comments. We have condensed and reorganized the introduction section, which is not shown here because of its length but is marked in blue in the revised manuscript. Briefly, we streamlined the discussion on the oxidative degradation of organic matter by atmospheric microorganisms, focusing on the most pertinent findings. Specifically, we removed the section discussing the link between atmospheric microorganisms and inorganic oxidants, as it was less central to the study's scope. Additionally, we reorganized the section on the importance of metabolomic studies, placing greater emphasis on findings directly relevant to atmospheric microbial metabolites. These revisions aim to enhance clarity and ensure the introduction is more concise while providing a solid foundation for the research presented. The revisions are marked blue in the revised manuscript in Page 2, Line 52 – 72 and Line 83 – 96.

R1 Comment 3: Lines 150-160: why the culture process were repeated for three cycles? What is the rationale behind this? Bacteria were culture at 37 °C and fungi were cultured at 28 °C. Why these two temperatures were selected? will the culture temperature affect the exometabolome?

**Response:** In the isolation experiments, we checked the stability of each strain by repeating the culture process through multiple cycles, ensuring consistency and reproducibility of the organisms. Typically, environmental microbial researchers conduct microbial isolation by culturing at least three cycles in most studies (Dunbar et al., 2015; Timm et al., 2020; Yan et al., 2021). In our present study, we achieved stable and robust growth of the isolated microbial strains after performing three or four streaking cycles. However, when encountering poor growth or inconsistent single colonies during strain isolation, we can conduct additional repetitions to ensure the acquisition of pure single colonies.

In the revision, we added more details on the microbial cultivation and isolation processes in Line 134 – 138, Page 5.

"From each plate, all phenotypically distinct colonies were picked up and smeared onto fresh media for isolation. Single colonies were picked and re-streaked from each plate at least three times to isolate individual strains (Timm et al., 2020). Multiple streaking isolations are essential to obtain stable and pure strains by gradually diluting the microbial population and allowing for the isolation of individual colonies (Yan et al., 2021; Dunbar et al., 2015)."

In this part of the method section, we reference the following literatures:

Dunbar, D., Shade, D., Correa, A., Farina, J., Terry, K., Broomell, H., Graves, L., Chudoff, D., Schoff, C., and Cross, T.: An Optimized Enrichment Technique for the Isolation of *Arthrobacter* Bacteriophage Species from Soil Sample Isolates, Journal of Visualized Experiments, e52781, https://doi.org/10.3791/52781, 2015.

Timm, C. M., Loomis, K., Stone, W., Mehoke, T., Brensinger, B., Pellicore, M., Staniczenko, P. P. A., Charles, C., Nayak, S., and Karig, D. K.: Isolation and characterization of diverse microbial representatives from the human skin microbiome, Microbiome, 8, 58, https://doi.org/10.1186/s40168-020-00831-y, 2020.

Yan, D., Zhang, T., Bai, J.-L., Su, J., Zhao, L.-L., Wang, H., Fang, X.-M., Zhang, Y.-Q., Liu, H.-Y., and Yu, L.-Y.: Isolation, Characterization, and Antimicrobial Activity of Bacterial and Fungal Representatives Associated With Particulate Matter During Haze and Non-haze Days, Frontiers in Microbiology, 12, 793037, https://doi.org/10.3389/fmicb.2021.793037, 2021.

The temperatures used for culturing bacteria and fungi are typically chosen based on their optimum growth condition and are often detailed in the sections of materials and methods in literatures (Wang et al., 2023; Zhan et al., 2024; Liu et al., 2024; Yang et al., 2022; Yang et al., 2021).

The optimal temperature for culturing atmospheric bacteria depends largely on the specific species or strains involved. While temperatures between 30°C and 37°C are commonly used for a wide range of bacterial cultures, 37°C is frequently employed, particularly for species that are adapted to environments influenced by human activities. This temperature reflects conditions where human-associated microbial communities

thrive. In studies focused on airborne bacterial cultures, 37°C is often the preferred incubation temperature, as it is widely used in the isolation and count of bacteria from various environments, including caves, hospitals, schools, and sewage treatment plants (Tomazin et al., 2024; Jeong et al., 2022; Zhang et al., 2023).

On the other hand, many fungi, such as those found in soils or on decaying organic matter, prefer environments with moderate temperatures, typically ranging from 20°C to 30°C. Specifically, 28°C is often used in fungal cultures because it is slightly higher than room temperature and reflects the conditions many fungi encounter in their natural environments, supporting robust growth and replication for a wide range of fungal species (Riccardi et al., 2021; Wang et al., 2023).

To some extent, different temperatures affect the growth rate, metabolic activity, and overall behavior of microorganisms. In our study, we specifically focused on assessing the potential metabolic capacity of atmospheric microorganisms under optimal conditions. To achieve this, we carefully selected suitable temperatures to ensure we obtained the desired results.

In this part of the method section, we reference the following literatures:

Liu, Y., Deng, G., Liu, H., Chen, P., Pan, Y., Chen, L., Chen, H., and Zhang, G.: Seasonal variations of airborne microbial diversity in waste transfer stations and preventive effect on *Streptococcus pneumoniae* induced pulmonary inflammation, Science of The Total Environment, 912, https://doi.org/10.1016/j.scitotenv.2023.168888, 2024.

Wang, S., Qian, H., Sun, Z., Cao, G., Ding, P., and Zheng, X.: Comparison of airborne bacteria and fungi in different built environments in selected cities in five climate zones of China, Science of The Total Environment, 860, 160445, https://doi.org/10.1016/j.scitotenv.2022.160445, 2023.

Yang, L., Shen, Z., Wei, J., Wang, X., Xu, H., Sun, J., Wang, Q., and Cao, J.: Size distribution, community composition, and influencing factors of bioaerosols on haze and non-haze days in a megacity in Northwest China, Science of The Total Environment, 838, 155969, https://doi.org/10.1016/j.scitotenv.2022.155969, 2022.

Yang, L., Shen, Z., Wang, D., Wei, J., Wang, X., Sun, J., Xu, H., and Cao, J.: Diurnal Variations of Size-Resolved Bioaerosols During Autumn and Winter Over a Semi-Arid Megacity in Northwest China,

Geohealth, 5, e2021GH000411, https://doi.org/10.1029/2021GH000411, 2021.

Zhan, J., Xu, S., Zhu, Y., Han, Y., Li, L., Liu, J., and Guo, X.: Potential pathogenic microorganisms in rural wastewater treatment process: Succession characteristics, concentration variation, source exploration, and risk assessment, Water Research, 254, https://doi.org/10.1016/j.watres.2024.121359, 2024.

R1 Comment 4: Lines 223: what does this word "resistant" mean here?

**Response:** Thank you for your comments. The word "resistant" means peaks that appear in both the initial medium and the final culture solution. We mentioned the definition in Line 234, Page 8. These molecules are not entirely consumed by the microorganisms, although may be partially produced but cannot be distinguished as exometabolites. Similar terminology "resistant" is also used in previous literatures (Chen et al., 2023; Yu et al., 2019; Mitra et al., 2013).

Chen, H., Uzun, H., Tolić, N., Chu, R., Karanfil, T., and Chow, A. T.: Molecular Transformation of Dissolved Organic Matter during the Processes of Wildfire, Alum Coagulation, and Disinfection Using ESI(−) and ESI(+) FT-ICR MS, ACS ES&T Water, 3, 2571-2580, https://doi.org/10.1021/acsestwater.3c00135, 2023.

Mitra, S., Wozniak, A. S., Miller, R., Hatcher, P. G., Buonassissi, C., and Brown, M.: Multiproxy probing of rainwater dissolved organic matter (DOM) composition in coastal storms as a function of trajectory, Marine Chemistry, 154, 67-76, https://doi.org/10.1016/j.marchem.2013.05.013, 2013.

Yu, Z., Liu, X., Chen, C., Liao, H., Chen, Z., and Zhou, S.: Molecular insights into the transformation of dissolved organic matter during hyperthermophilic composting using ESI FT-ICR MS, Bioresource Technology, 292, 122007, https://doi.org/10.1016/j.biortech.2019.122007, 2019.

R1 Comment 5: Figure 2 panel b: it would be better to label each column. Otherwise, it is not clear which bar is the consumed, resistant, and produced organic matter.

**Response:** We have modified Figure 2 panel b by labeling the abbreviations of Consumed, Resistant, and Produced as C, R, and P below each column and adding a legend. All abbreviations are explained in the figure caption.

"Figure 2. Changes in molecular compositions of culture media during the metabolic processes of six typical atmospheric culturable bacteria. (a) Van Krevelen diagrams illustrate changes in different categories of organics in the media after incubation. The top three Van Krevelen diagrams represent the chemical composition of Gram-positive bacteria, the bottom three Van Krevelen diagrams correspond to Gram-negative bacteria. (b) The stacked bar charts indicate the formula numbers of different categories of consumed (labeled as C), resistant (R), and produced (P) organic matter."

R1 Comment 6: Figure 4 pane a: I assume that the two pie charts represent the distribution of number concentration and intensity concentration, respectively. if yes, it might be better to use different color scales for these two charts to differentiate them. Similar suggestion for figure 5a.

**Response:** Thank you for your comments. The two pie charts in Figure 4a and Figure 6a (previously referred to as Figure 5a in the original manuscript) represent the number and intensity of molecules composed of different elements (i.e., CHO, CHON, CHOS, CHONS, and CHONSP). We have rearranged Figures 4 and 6 for improved readability and clarity, and optimized the colors and sizes of the legends. The adjustments to the figure captions are highlighted in blue in the revised manuscript.

Thank you very much for your comments and suggestions. Your any further comments and suggestions are appreciated.

**Response to Referee 2 (Prof. Maki Teruya):**

**General comments**

R2 Comment 1: This paper reported on the exometabolite (extracellular chemical components) produced by microbial cultures, which are isolated from urban aerosol samples. Investigating the extracellular components from airborne microorganisms are important for understanding the nutrient input by deposition, the survival of airborne microorganisms, the microbial growth, the chemical cycles, the pathogenic strategy and so on. However, I think this paper just showed the measurement data of exometabolite discussing no categorization of ecological means for each isolates producing exometabolite. Additionally, although there are several species of isolates, the exometabolite producing by each species are not categorized and the ecological characteristics of exometabolite have not discussed clearly.

I think this manuscript could be accepted after redrafting the manuscript.

**Response:**

Dear Prof. Maki Teruya,

Thank you very much for your thorough review and valuable comments on our manuscript. We have carefully considered your suggestions and made corresponding revisions to both the main text and the supplemental information to improve the quality of the manuscript.

We acknowledge your concerns regarding the lack of species-specific categorization of exometabolites and the need for a more explicit discussion of their ecological implications. In response, we have revised the manuscript to provide a more detailed categorization of the exometabolites produced by each isolated species and expanded our discussion on their potential ecological roles. We are confident that these revisions enhance the clarity and depth of the manuscript.

The detailed responses can refer to the point-to-point responses below. Here are the brief responses to your comments.

1) We have classified the bacterial strains into Gram-negative and Gram-positive based on their cellular properties, and the fungal strains into yeasts and molds based on

their growth mode. As a results, all diagrams in the manuscript have been updated to reflect this classification.

2) Addtionally, we have restructured the manuscript to focus on two main aspects: (a) the characterization of molecular composition of the exometabolome for different microbial strains and (b) a discussion on the possible ecological roles of these exometabolites.

**Some major comments:**

R2 Comment 2: I think the total twelve strains are small for discussing the contribution of organic materials to carborne cycles in atmosphere, because there are many kinds of airborne microorganisms, which are unculturable (>90% of microorganisms cannot be cultured but they are living). The authors should discuss deeply on microbial compositions to focus on the dominant species, which can be focused as keystone species.

**Response:** We acknowledge that a large proportion (over 90%) of airborne microorganisms are unculturable. However, in this study, the bacterial and fungal strains selected in this study represent major taxa commonly found in the atmosphere (refer to Section 3.1). In this presented study, we focused on the dominant culturable species as a first step in understanding their potential contribution to organic material cycling in atmospheric biogeochemical cycle.

In addition to this, we also identified the dominant microbial taxa in aerosols collected at the same sampling site through 16S rRNA high-throughput sequencing and metagenomic analysis (Fig. R1). A detailed characterization of the microbial composition from the same site will be discussed in our further studies.

As shown in Fig. R1, at the genus level, the dominant fungi included *Talaromyces*, *Aspergillus*, and *Penicillium*, which were also cultured and isolated in this study. Similarly, the dominant bacterial genera included *Streptomyces*, *Pantoea*, and *Pseudomonas* (Fig. R1a). At the species level, *Penicillium oxalicum*, *Aspergillus niger*, and *Talaromyces* sp. were among the most abundant taxa, with their relative abundance

ranking within the top 0.5% of all annotated species (Fig. R1b). Additionally, species including *Rhodotorula mucilaginosa*, *Pantoea vagans*, *Bacillus subtilis*, and *Streptomyces pratensis* were also identified, with relative abundances placing them in the top 10% of all detected species. Therefore, the microorganisms selected for our exometabolomic studies indeed represent the dominant atmospheric taxa, ensuring that our findings are relevant to the most prevalent species.

In response to your suggestion, we expanded the discussion to address the broader microbial community, emphasizing the potential ecological roles of the dominant species and their relevance to atmospheric processes. The revised manuscript now reflects these considerations.

Page 7, Line 213 – 217: "The isolated bacterial genera represent major taxa commonly found in the atmosphere (Yan et al., 2021; Lee et al., 2017; Calderon-Ezquerro et al., 2021). Gram-positive bacteria such as *Bacillus* and *Streptomyces* possess cell walls predominantly composed of N-acetylmuramic acid. While, Gram-negative bacteria such as *Pantoea*, *Erwinia*, *Stenotrophomonas*, and *Pseudomonas* have cell walls rich in lipopolysaccharides (LPS)/endotoxin (Ruiz-Gil et al., 2020)."

Page 13, Line 408 – 418: "Most bacterial genera cultured from the atmosphere in various worldwide investigations are Gram-positive bacteria, with *Bacillus* as one of the most abundant genera (Yoo et al., 2019; Maki et al., 2010; Hua et al., 2007). The *Bacillus* genus has often been identified as a dominant taxon in dust aerosols (Rossi et al., 2024; Péguilhan et al., 2023; Maki et al., 2022). For instance, *B. subtilis* has been found to increase significantly during dust events and maintain culturability. *B. subtilis* is known for its high adaptability to diverse extreme conditions, e.g., salt- and temperature-tolerant, allowing it to survive in a wide range of environments (Losick, 2020; Liu et al., 2018; Maki et al., 2022). The versatile metabolic capacity may play a physiological role in facilitating bacterial survival under harsh environments. Previous studies demonstrated that *Bacillus* sp. 3B6 was able to efficiently biotransform sugars into extracellular polymeric substances to protect this bacterium under hostile environment conditions, including cold temperature, radical exposure, and freeze

processes (Matulová et al., 2014). Additionally, members of the *Bacillus* genus are known for their ability to form endospores, enabling them to withstand atmospheric stressors and survive long-range transport (Nicholson et al., 2000)."

Page 14, Line 419 – 427: "*Pantoea* is a highly diverse genus found in both aquatic and terrestrial environments (Walterson et al., 2015; Murillo-Roos et al., 2022). *Pantoea* species exhibit a robust oxidative stress response by synthesizing enzymes such as catalase, superoxide dismutase, and peroxidase (Dahiya et al., 2024; Tambong, 2019). *Pseudomonas* species are well-adapted to atmospheric conditions, and they often oxidize common atmospheric volatile organic compounds, including formaldehyde and methanol (Husarova et al., 2011). *Streptomyces* is a dominant genus in aerosols from various megacities and background areas, especially in winter (Chen et al., 2021a; Petroselli et al., 2021; Li et al., 2019). Furthermore, *Streptomyces* has been widely reported in aerosols from European deserts and urban sewage treatment plants (Núñez et al., 2024; Zhan et al., 2024). *Streptomyces* is a drought-tolerant bacterial genus with spore-forming ability (Taketani et al., 2016). Their spores can enter a stable and quiescent state under environmental stress, enabling survival in diverse natural conditions (Naylor et al., 2017)."

[Figure]

**Figure R1** Characterization of atmospheric microbial composition in urban Tianjin. (a) Microbial composition at the genus level with the transcript per kilobase per million mapped reads (TPM) greater than 1000 reads; (b) Microbial composition at the species level with the TPM greater than 500 reads (top 0.5%) (unpublished data).

R2 Comment 3: Extracellular products are composed of several kinds of chemical components, which contribute to the roles of airborne microorganisms. Accordingly, after the categories of ecological merits by the chemical components are defined, the categories of chemical components are established for understanding the roles of extracellular products on airborne bacterial ecology. For examples, there are the some categories such as 1) Survivals of airborne microorganisms in atmosphere, 2) Microbial growth in atmosphere and after deposition, 3) Chemical cycles in several environments, and 4) Pathogenic abilities using some components and so on.

**Response:** Thank you for your insightful comments. We fully agree that categorizing the ecological roles of extracellular products based on their chemical components is crucial for a deeper understanding of airborne microorganisms' ecology. In response, we have revised the manuscript to better define the ecological significance of the chemical components produced by the isolated microorganisms. Importantly, we have highlighted certain interesting products and discussed their potential ecological functions, mainly in the Results or Discussion sections.

These revisions are integrated throughout several paragraphs in the manuscript. All changes are clearly marked in blue in the revised manuscript.

The main revisions include:

1) Page 14, Line 429 – 436: "Specifically, the exometabolites of *Penicillium* and *Talaromyces* were enriched with pigment compounds, particularly CRAM-like molecules with CHO elemental compositions (Fig. 6a and 6b). Some aliphatic/peptide-like and aromatic-like compounds were biotransformed into CRAM-like pigment molecules ($C_{14}H_{10}O_6$, $C_{23}H_{24}O_7$, $C_{26}H_{29}NO_8$, $C_{14}H_{10}O_6$, and $C_{15}H_{10}O_5$) (Fig. 7c and 7d). Microbial pigments are chemically diverse, including

flavonoids, isoprenoids, porphyrins, N-heterocyclics, and polyketides (Contreras-Machuca et al., 2022; Morales-Oyervides et al., 2020; Venkatachalam et al., 2018). Recent studies have highlighted the potential of *Penicillium* and *Talaromyces* as robust producers of natural pigments, noted for their thermal, pH, and light stability (Ugwu et al., 2021; Akilandeswari and Pradeep, 2016)."

2) Page 15, Line 481 – 487: "A large number of CHONS compounds, belonging to aliphatic/peptide-like molecules, were detected in the metabolites of the Gram-negative bacteria *Stenotrophomonas* sp. and *P. baetica*, accounting for 40.6% (55.3% of intensity) and 33.2% (61.8% of intensity), respectively (Fig. 4b and S5). These compounds, including extracellular polymeric substances, play a crucial role in microbial survival by forming protective biofilms and contributing to the atmospheric biogeochemical cycle. The most frequently detected genus in the atmosphere, *Pseudomonas*, is a key producer of biosurfactants, which can impact atmospheric chemistry (e.g., secondary organic formation) and modify cloud microphysics by enhancing cloud condensation nuclei activation owing to their exceptional ability to reduce surface tension (Delort et al., 2010)."

3) Page 16, Line 488 – 496: "Building on prior studies, some *Stenotrophomonas* spp. are known to synthesize a range of biologically active compounds, including a variety of antibiotic enzymes such as β-lactams, aminoglycosides, and macrolides, alongside chitinases, lipases, and proteases (Ryan et al., 2009; Peleg and Abbott, 2015; Wang et al., 2018). These findings suggest that many of the CHONS compounds identified in the exometabolites are likely proteases or their degradation products, underscoring the pivotal role of *Stenotrophomonas* in protein turnover and nutrient cycles in the land/water-air interfaces (Liu et al., 2016). Furthermore, the molecular composition of the exometabolites pointed to an enrichment of amino acid metabolic pathways, like tyrosine metabolism (KO00350) and histidine metabolism (KO00340) (Fig. 8a). These pathways, coupled with the known reactivity of amino acids, highlight the importance of considering the amino acid presence and transformation when investigating the chemical composition and

oxidative processes occurring within cloud water (Bianco et al., 2016)."

4) Page 16, Line 502 – 510: "*R. mucilaginosa* was predominantly involved in carbohydrate metabolism (Fig. 8b). In this study, several saturated fatty acids, including myristic acid ($C_{14}H_{28}O_2$), stearic acid ($C_{18}H_{36}O_2$), and palmitic acid ($C_{16}H_{32}O_2$), and unsaturated fatty acids including linoleic acid ($C_{18}H_{32}O_2$) and linolenic acid ($C_{18}H_{30}O_2$), were detected. These organic acids were also abundantly present in atmospheric aerosols and may affect the atmospheric chemical and physical processes, e.g., cloud formation (Mkoma and Kawamura, 2013; Balducci and Cecinato, 2010; Raymond and Pandis, 2002). Notably, palmitic acid, stearic acid, and linoleic acid are potential molecular tracers for estimating the contribution of cooking emissions to organic aerosols (Cheng and Yu, 2020; Ma et al., 2023). However, our findings reveal that fatty acids were also abundant in the metabolites of *Rhodotorula*, suggesting that microorganisms may contribute to atmospheric fatty acid levels. This highlights the need to consider microbial sources when assessing the composition of atmospheric organic aerosols."

5) Page 16, Line 511 – 524: "Among airborne fungi, the genera *Aspergillus*, *Penicillium*, and *Talaromyces*, which are classified as mold fungi, are acknowledged as some of the most chemically prolific organisms. These genera can synthesize a wide variety of secondary metabolites (exometabolites) (Frisvad, 2015; Zhai et al., 2016; Morales-Oyervides et al., 2020). They produce specialized compounds such as polyketides, non-ribosomal peptides, and terpenes, further showcasing their metabolic versatility and biochemical complexity (Frisvad, 2015; Adelusi et al., 2022). This remarkable chemodiversity highlights their ecological significance in atmospheric biogeochemistry. *A. niger* is ubiquitous in the environment and plays a significant role in the global carbon cycle (Baker, 2006; Schuster et al., 2002; Šimonovičová et al., 2021). It is rich in genetic and metabolic diversity, acting as one of the most important microorganisms used in biotechnology (Sun et al., 2007; Jørgensen et al., 2011). In its natural growth state, *A. niger* possesses large cryptic biosynthetic gene clusters (BGCs), which

synthesize a wide range of extracellular enzymes to degrade special biopolymers in the environment, thus allowing the fungus to obtain nutrients (Yu et al., 2021; Romsdahl and Wang, 2019). Anthraquinones like $C_{15}H_{12}O_5$, $C_{15}H_{10}O_5$, and $C_{15}H_{10}O_6$ were detected in the product of *Penicillium* and *Talaromyces*, and this compound can protect symbiotic plants from pests or pathogens (Etalo et al., 2018). This metabolic plasticity underscores their ecological importance and the potential for biotechnological applications in various industries."

R2 Comment 4: The six isolates of bacteria or fungi can be classified to some groups in dependence on the taxon. The chemical components in extracellular products are discussed for each category to understand the ecological characteristics of each categorized microorganisms. This comments are relating to Comment 1.

**Response:** We revised the manuscript to include a more detailed categorization of the exometabolites produced by each isolated species and expanded the discussion on their potential ecological roles. We have classified bacterial strains into Gram-negative and Gram-positive based on their cellular properties, and fungal strains into yeasts and molds based on their growth modes. These revisions are all marked in blue in the revised manuscript.

The main revisions include:

1) Page 9, Line 285 – 293: "*Stenotrophomonas* sp. and *P. baetica*, both Gram-negative bacteria, metabolized and synthesized large quantities of CHONS compounds (55.3–61.8% of intensity), largely differed from the other four bacteria strains (only 3.5–21.5% of intensity). Notably, in exometabolites from *Stenotrophomonas* sp., CHONS compounds dominated the lipid-like and aliphatic/peptide-like molecules, distinguishing it from the other bacterial strains (Fig. 4b). Microbes can convert bioavailable lipid-like and protein-like nitrogenous compounds into more oxygenated, unsaturated (more refractory) nitrogenous CRAM/lignin-like compounds (Osborne et al., 2013). Additionally, *Stenotrophomonas* sp. produced a higher number (903 formulas) of high molecular weight compounds ($m/z$ >500)

(Table S6), likely representing extracellular polymeric substances, e.g., polysaccharides, extracellular enzymes, and cellular debris (Vandana et al., 2023; Moradali and Rehm, 2020)."

2) Page 10, Line 305 – 311: "For Gram-positive bacteria, the elemental compositions of the unique formulas were similar, with CHON compounds accounting for more than 50% of all compounds (Fig. 5a). In contrast, for Gram-negative bacteria, CHON compounds were predominant (48.1%) in the unique formulas specific to *P. vagans*, and CHONS compounds are predominantly those specific to *Stenotrophomonas* sp. and *P. baetica*, with proportions of 70.2% and 71.4%, respectively (Fig. 5a). In addition, CHO compounds comprised over 25% of the unique formulas found exclusively in *P. vagans* (Fig. 5a). Notably, among these unique CHO compounds, the most abundant formulas belonged to the $O_9$ class, dominated by $C_{18}H_{28}O_9$ and $C_{13}H_{16}O_9$, indicating higher oxidation states (Fig. S5c)."

3) Page 10, Line 319 – 324: "Unlike the bacterial strains, which all produced high fractions of CHONS compounds, CHONS compounds appeared only in exometabolites of *R. mucilaginosa*, a yeast species, accounting for approximately 19% of the total formula number (Fig. 6a). This indicates a diversity of sulfur-related metabolic processes for *Rhodotorula*. In contrast, the exometabolites produced by mold fungi, specifically from the genera *Penicillium* (*Penicillium oxalicum* and *Penicillium aurantiogriseum*) and *Talaromyces* (*Talaromyces* sp.), were predominantly CHO compounds, accounting for over 50% of the total exometabolite intensity (Fig. 6a)."

4) Page 15, Line 481 – 487: "A large number of CHONS compounds, belonging to aliphatic/peptide-like molecules, were detected in the metabolites of the Gram-negative bacteria *Stenotrophomonas* sp. and *P. baetica*, accounting for 40.6% (55.3% of intensity) and 33.2% (61.8% of intensity), respectively (Fig. 4b and S5). These compounds, including extracellular polymeric substances, play a crucial role in microbial survival by forming protective biofilms and contributing to the atmospheric biogeochemical cycle. The most frequently detected genus in the

atmosphere, *Pseudomonas*, is a key producer of biosurfactants, which can impact atmospheric chemistry (e.g., secondary organic formation) and modify cloud microphysics by enhancing cloud condensation nuclei activation owing to their exceptional ability to reduce surface tension (Delort et al., 2010)."

R2 Comment 5: The Results and Discussion should be separated each, because there are many overlapped parts and redundancy, which make readers understand this hardly.

**Response:** Thank you for your valuable comments. To enhance the readability of the paper, we have separated and reorganized the Results and Discussion sections according to the specific findings and conclusions. All corrections are marked in the revised manuscript.

R2 Comment 6: I think the authors are not familiar with biology. So some biologist suggestion is needed for categorizing the taxon and describing the taxon manes.

**Response:** Thank you for your comments. We are not sure whether do you mean we should designate the name of microbial strains by using letters, symbols, or numbers, and use the new names of bacterial phyla standardized in the year of 2021. For the former one, we renamed the strains using TJB1−24 and TJF1−16 in Table S2, S3, S4. But for concision, we use species names instead of strains names in the main text and figures. For the latter one, we have checked and used the new taxon names in the manuscript and Figure 1. The corrections are marked in blue in the revised manuscript.

**Some minor comments:**

R2 Comment 7: L19: What is CHON?

**Response:** According to the elemental composition of organic compounds, their molecular composition is classified into four categories: CHO, CHON, CHOS, and CHONS. This classification has been commonly used in mass spectrometry analyses. For example, CHON means that the organic compounds contain only four elements: carbon, hydrogen, oxygen, and nitrogen without any other elements.

In the revision, we added "For example, CHO indicates that the organic compounds in this category contain only three elements: carbon, hydrogen, oxygen, with no other elements." in Line 198 – 199, Page 7.

R2 Comment 8: L22: Fungi also produce amino acid.

**Response:** We complemented this sentence by integrating the results presented in Fig. 8 (b). The revision has been made and is marked in blue (see the annotated revision).

Page 1, Line 24 – 25: "Bacteria exhibited proficiency in amino acid synthesis, while fungi were actively involved in amino acid metabolism, transcription, and expression processes."

R2 Comment 9: L25-L28: This is general summary. Please summarize specific topics for this study.

**Response:** We understand the importance of providing a more focused summary highlighting the specific topics of our study. We revised the summary to emphasize the main objectives and findings of our study, which will be more explicitly detailed in the revised manuscript to provide a clearer understanding of the study's scope and contributions.

Page 1, Line 27 – 29: "This study provides new insights into the transformation and potential oxidative capacity of atmospheric microorganisms concerning organic matter at air-land/water interfaces. These findings are pivotal for assessing the biogeochemical impacts of atmospheric microorganisms following their deposition."

R2 Comment 10: L30-L40: The sampling is performed in Asian site. The papers relating to Asian-dust transport of bioaerosols are needed. I think European bioaerosols are different from Asian bioaerosols. So the author can isolate the *Bacillus* strain efficiently.

**Response:** Thank you for your comments. We acknowledge the importance of including literature on the transport of bioaerosols associated with Asian dust events.

We have incorporated relevant references to provide a more comprehensive context. Regarding the comparison between European and Asian bioaerosols, we agree that geographical differences may influence microbial composition. However, *Bacillus* is also illustrated as the dominant bacterial genus in many studies on the composition of European bioaerosol bacterial communities (Rossi et al., 2024; Péguilhan et al., 2023; Chatoutsidou et al., 2023; Nunez et al., 2021). For example, the relative abundance of *Bacillus* was even 13% among atmospheric bacteria at the Puy de Dôme observatory in France. Our study focused on dominant taxa found in the atmospheric samples, with *Bacillus* being one of the primary subjects. We appreciate your suggestion and have incorporated a discussion on the specificity of bioaerosols across different regions into the revised manuscript.

1) Page 13, Line 408 – 418: "Most bacterial genera cultured from the atmosphere in various worldwide investigations are Gram-positive bacteria, with *Bacillus* as one of the most abundant genera (Yoo et al., 2019; Maki et al., 2010; Hua et al., 2007). The *Bacillus* genus has often been identified as a dominant taxon in dust aerosols (Rossi et al., 2024; Péguilhan et al., 2023; Maki et al., 2022). For instance, *B. subtilis* has been found to increase significantly during dust events and maintain culturability. *B. subtilis* is known for its high adaptability to diverse extreme conditions, e.g., salt- and temperature-tolerant, allowing it to survive in a wide range of environments (Losick, 2020; Liu et al., 2018; Maki et al., 2022). The versatile metabolic capacity may play a physiological role in facilitating bacterial survival under harsh environments. Previous studies demonstrated that *Bacillus* sp. 3B6 was able to efficiently biotransform sugars into extracellular polymeric substances to protect this bacterium under hostile environment conditions, including cold temperature, radical exposure, and freeze processes (Matulová et al., 2014). Additionally, members of the *Bacillus* genus are known for their ability to form endospores, enabling them to withstand atmospheric stressors and survive long-range transport (Nicholson et al., 2000)."

2) Page 14, Line 423 – 427: "*Streptomyces* is a dominant genus in aerosols from

various megacities and background areas, especially in winter (Chen et al., 2021a; Petroselli et al., 2021; Li et al., 2019). Furthermore, *Streptomyces* has been widely reported in aerosols from European deserts and urban sewage treatment plants (Núñez et al., 2024; Zhan et al., 2024). *Streptomyces* is a drought-tolerant bacterial genus with spore-forming ability (Taketani et al., 2016). Their spores can enter a stable and quiescent state under environmental stress, enabling survival in diverse natural conditions (Naylor et al., 2017)."


**Response:** Thank you for your comments. This study only provides a preliminary result on the potential transformation and potential oxidative capacity of culturable urban atmospheric microorganisms for organic matter based on typical microbial strains, without considering different types of aerosols.

During the sampling period in this study, Asian dust events were not recorded. While, we are utilizing exometabolome assays and data analysis methods from this study to investigate the activity and metabolic capacity of atmospheric microorganisms following the deposition of Asian dust into surface seawater. We have conducted a thorough examination of these microorganisms' responses and are currently compiling and analyzing the specific findings. The related results will be published in future work.

R2 Comment 13: L91-L101: The analytical methods should be the results relating to more ecological topics. I think this section just show analytical method index.

**Response:** Thank you for your comments. We understand your concern regarding the need for the analytical methods to be more closely related to ecological topics. In this section, we intend to present the analytical methods and demonstrate their validity, which is why it may appear more focused on methodology. We have condensed this section and added some applications in ecology.

Page 3, Line 85 – 96:

"Microbial exometabolites hold great ecological significance, as they can serve as substrates for other microorganisms, facilitating population growth (Douglas, 2020). For instance, the metabolome of *Pseudomonas graminis* exposed to oxidative stress involves carbohydrate, glutathione, energy, lipid, peptide, and amino acid metabolism pathways (Wirgot et al., 2019). Furthermore, microbial exometabolites, including extracellular polymeric substances and pigments, are vital for the survival of atmospheric microorganisms, enabling them to withstand long-range transport and extreme conditions in the atmosphere or at the ocean surface (Erkorkmaz et al., 2023;

Bryan et al., 2019). Recently, a framework for atmo-ecometabolomics, from sampling to data analysis, was developed to characterize the molecular composition of aerosols (Rivas-Ubach et al., 2019). This framework also provides insights into how aerosol chemical compositions impact ecosystem structure, function, and biogeochemistry. However, previous studies have not fully explored the role of metabolically active microorganisms in atmospheric aerosols. Deepening our understanding of the metabolome of atmospheric microorganisms is essential for predicting their ecological roles in biogeochemical cycles and assessing their broader impacts on ecosystem functions and atmospheric processes."

R2 Comment 14: L110: I do not think KEGG analysis is needed for this paper. However, if KEGG analysis is remained, the KEGG analysis should be introduce in the introduction section.

**Response:** Thank you for your comments. We have now incorporated the KEGG explanation into the introduction section, with these revisions marked in blue in the revised manuscript. We chose to use KEGG analysis because it allows us to comprehensively explore the biochemical reaction processes that are highly relevant to atmospheric microorganisms. The metabolome provides a dynamic snapshot of an organism's metabolic state, capturing the complex interactions among genes, proteins, and other factors. By utilizing KEGG identifiers for the metabolites we identified, we could map these against bioinformatic databases using the KEGG mapper in batch mode, systematically identifying and analyzing the involved biochemical pathways. This approach offers a robust framework for understanding the complex metabolic functions and key shifts in microbial processes, helping to elucidate the unique metabolic processes that support microbial survival and adaptation in atmospheric environments.

Page 4, Line 101 – 108:

"Relevant biosynthetic pathways can be traced through chemical information, relying on the Kyoto Encyclopedia of Genes and Genomes (KEGG) database (Lopez-

Ibañez et al., 2021). KEGG is a comprehensive database resource designed to facilitate interpreting high-level functions and systems by integrating molecular-level data, including cellular, organismal, and ecological processes. This database is particularly valuable for analyzing large-scale molecular datasets derived from genomic sequencing and other high-throughput technologies, enabling the exploration of complex biological pathways and networks (Kanehisa et al., 2017). Biochemical pathways were identified by mapping the molecular composition of metabolites obtained by FT-ICR MS to chemical structures in the KEGG COMPOUND Database using the analytical pipeline developed by Ayala-Ortiz et al. (2023)."

R2 Comment 15: L123: How many samples are collected? The sample numbers should be describe and the samples for each isolates are explained. Additionally, the environmental factors have to be discussed when the isolates are obtained.

**Response:** Thank you for your thoughtful suggestion. Regarding the experimental design, we conducted our experiment with three samples, each analyzed in parallel with three replicates. While we acknowledge that the sample size may be somewhat limited, we carefully distinguished different microbial phenotypes during the isolation process. Notably, the microorganisms selected for our exometabolomic studies represent the dominant taxa of atmospheric microorganisms, ensuring that our findings are relevant to the most prevalent species. Additionally, we would like to clarify that the primary aim of our research was to investigate the potential metabolic capacities of these atmospheric microorganisms, rather than the influence of environmental factors. Nonetheless, we highly value your suggestions and will certainly incorporate these considerations into future studies. We have added relevant notes about the samples to the revised manuscript.

Page 5, Line 129 – 134:

"Each sample underwent three parallel experiments. Microorganisms on the filters were detached using low-power ultrasonication in an ice bath for 5 min and then centrifuged at 250 rpm for 30 min. Then, a 100 μL aliquot of the well-mixed suspension

was spread-plated onto the two types of solid media. For each suspension, three independent replicates were prepared to ensure that both media were thoroughly coated and evaluated. The plates were incubated at a constant temperature of 37°C for 48 h for bacteria, and at 28°C for 72 h for fungi, with daily growth monitoring (Wang et al., 2023; Timm et al., 2020)."

R2 Comment 16: L184-L185: The chemical components should be defined in detail, because these are familiar with the readers for ACP.

**Response:** We added the meanings of these chemical components and labeled them in blue.

Page 7, Line 198 – 199:

"For example, CHO indicates that the organic compounds in this category contain only three elements: carbon, hydrogen, and oxygen, with no other elements."

R2 Comment 17: L185: KEGG explanation is needed in the Introduction. Why was this used for this paper?

**Response:** Thank you for your comments. We have now incorporated the KEGG explanation into the introduction section, with these revisions marked in blue in the revised manuscript. We chose to use KEGG analysis because it allows us to comprehensively explore the biochemical reaction processes that are highly relevant to atmospheric microorganisms. The metabolome provides a dynamic snapshot of an organism's metabolic state, capturing the complex interactions among genes, proteins, and other factors. By utilizing KEGG identifiers for the metabolites we identified, we could map these against bioinformatic databases using the KEGG mapper in batch mode, systematically identifying and analyzing the involved biochemical pathways. This approach offers a robust framework for understanding the complex metabolic functions and key shifts in microbial processes, helping to elucidate the unique metabolic processes that support microbial survival and adaptation in atmospheric environments.

Page 4, Line 101 – 108:

"Relevant biosynthetic pathways can be traced through chemical information, relying on the Kyoto Encyclopedia of Genes and Genomes (KEGG) database (Lopez-Ibañez et al., 2021). KEGG is a comprehensive database resource designed to facilitate interpreting high-level functions and systems by integrating molecular-level data, including cellular, organismal, and ecological processes. This database is particularly valuable for analyzing large-scale molecular datasets derived from genomic sequencing and other high-throughput technologies, enabling the exploration of complex biological pathways and networks (Kanehisa et al., 2017). Biochemical pathways were identified by mapping the molecular composition of metabolites obtained by FT-ICR MS to chemical structures in the KEGG COMPOUND Database using the analytical pipeline developed by Ayala-Ortiz et al. (2023)."

R2 Comment 18: L193: As described, I recommend to separate to each section of Result and Discussion.

**Response:** To enhance the readability of the paper, we have separated and reorganized the Results and Discussion sections according to the specific findings and conclusions. All corrections are marked in the revised manuscript.

R2 Comment 19: L196-L197: The strain numbers should be described.

**Response:** We double-checked the content in Line 196 – 197 (in the original manuscript), which are part of the description of the isolation results, mainly describing the dominant taxa of culturable bacteria at the phylum and genus levels. It does not address specific strains or strain numbers.

R2 Comment 20: L262: The genus name should be initial after second using. For example, *A. niger*. This is biological rule.

**Response:** Thank you for your comments. We have corrected the microbial names following this rule throughout the manuscript. The corrections are labeled in blue.

R2 Comment 21: Fig.1. The new phyla name of bacteria should be used.

**Response:** Thank you for your comments. We modified the bacterial phylum names in Figure 1, as well as in the main text and supplement (Table S2), to the new names standardized in the year of 2021.

Page 7, Line 211 – 212: "The culturable bacterial community primarily consisted of Pseudomonadota (45.8%), Bacillota (37.5%), and Actinomycetota (16.7%)."

R2 Comment 22: L362-L430: This section is overlapped to Section 3.2. Please combine them.

**Response:** Thank you for your comments. We understand your concern regarding the potential overlap between these sections. However, the structure of our article is intentionally designed to first present the molecular characterization of the exometabolites (Sect. 3.2 and 3.3), followed by an analysis of the metabolic processes (Sect. 3.4). This sequence allows us to discuss the metabolic processes directly related to the identified products, ensuring a more coherent flow of information. By organizing the sections in this way, we align to corroborate the identified metabolites with the metabolic pathways responsible for their production, providing a more comprehensive understanding of the data. We have made the necessary changes in the manuscript and hope that this clarifies the structure of our article.

Thank you very much for your comments and suggestions. Your any further comments and suggestions are appreciated.

**Response to Referee 3:**

In this study, the authors examined the microbial exometabolism production of culturable airborne microorganisms from an urban atmosphere. A diverse range of products, including CHON, CHONS, and CHO compounds, were identified for both bacteria and fungi. Significant variations in metabolites were observed among different strains. In terms of amino acid synthesis, transcription and expression processes, lipid metabolism, amino acid metabolism, and carbohydrate metabolism, there was a wide variation among bacterial and fungal species. The results provide a comprehensive examination of metabolite characteristics at the molecular level for typical culturable airborne microorganisms. Overall, this work is novel and interesting, and the ideas and data presented in the manuscript are credible. This paper could be accepted after appropriately revision. More detailed comments/suggestions are provided below.

R3 Comment 1: L78 The influence of dry deposition dust on marine terrestrial ecological environments was discussed, with a particular focus on microbial communities. Additionally, the effects of wet deposition, including cloud, fog, rain, or snow processes should clarify.

**Response:** Thank you for your comments. We added the impact of wet deposition on marine ecosystems in the introduction section, which is marked in blue in the revised manuscript.

 Page 3, Line 73 – 80: "Wet deposition (e.g., rainfall) brings a wealth of nutrients and microorganisms to surface waters, increasing chlorophyll *a* concentration and enhancing carbon and nitrogen fixation. Short-term wet deposition events play an important role in the temporal variation of new nitrogen production and phytoplankton dynamics in the southeastern Mediterranean, as demonstrated by both experimental and in situ observations (Rahav et al., 2021). Additionally, one notable study demonstrated that rainwater influenced by Saharan dust significantly increased bacterial populations in high-altitude lake water, with rare taxa becoming dominant based on cultivation experiments (Peter et al., 2014). Despite these findings, there is a lack of research on

how atmospheric microorganisms metabolize and degrade organic matter, making it challenging to accurately assess their metabolic processes and potential impacts on atmospheric chemistry and biogeochemical cycles."

R3 Comment 2: Line 102-109, this section is an introduction to the FT-ICR MS method, which could be integrated into section 2. Materials and methods.

**Response:** We have reorganized the introduction section and moved the part introducing FT-ICR MS to the Materials and methods section. The revisions have been highlighted in blue in the revised manuscript.

Page 6, Line 184 – 185: "Since FT-ICR MS determines potent polarity molecules with molecular weight (MW) from 100 Da to 1000 Da, pigments or extracellular enzymes secreted by bacteria or fungi can be detected and identified."

R3 Comment 3: Line 123, the sampling period is three days, yet it remains unclear whether the sample volum satisfies the prerequisites for analysis. Could you elucidate the criteria employed to determine the sampling dates? Were these days being clean, polluted, or sunny or rainy/snowy days? Furthermore, could you elaborate on why the sampling time was specifically set between 15:00-19:00? Please offer further insights into the sampling procedure.

**Response:** Thank you for your comments. Our exploratory study specifically focused on culturable atmospheric microorganisms which are prevalent and highly active, as well as their potential effects on atmospheric biogeochemistry. Therefore, we did not extensively consider the effects of weather conditions on the isolation and cultivation of atmospheric microorganisms. The weather conditions and air quality during the sampling periods are summarized in Table R1, showing air quality levels that ranged from good, moderate, to lightly polluted, representing typical winter conditions.

Regarding the sampling time (15:00 – 19:00), we randomly selected this period to capture the dominant taxa prevalent among atmospheric microorganisms. Unfortunately, however, we could not document significant differences in isolates

among the three samples. This exploratory study has elucidated the metabolic activity of atmospheric microorganisms and the molecular composition of their exometabolites, providing a basis for investigating their potential role in biogeochemical cycles, particularly the carbon cycle.

We thank you for your suggestions and will consider this in our subsequent studies.

Table R1 Weather conditions and air quality during the sampling periods.

| Date | Temperature (°C) | Weather | AQI | Air quality level | PM$_{2.5}$ | PM$_{10}$ | SO$_2$ | NO$_2$ | CO | O$_3$ |
|---|---|---|---|---|---|---|---|---|---|---|
| 2022.01.05 | -4 – 4 | fine | 68 | Moderate | 39 | 71 | 11 | 54 | 1 | 29 |
| 2022.01.08 | -2 – 9 | fine | 125 | Lightly polluted | 95 | 129 | 7 | 53 | 1.2 | 48 |
| 2022.01.11 | -5 – 1 | cloudy | 40 | Good | 12 | 25 | 9 | 32 | 0.7 | 48 |

AQI: Air Quality Index; Unit of value: μg/m$^3$ (mg/m$^3$ for CO)

R3 Comment 4: Line 132, the methodology for microbial culture and separation must be detailed, including specific parameters such as centrifugal speed and the volume of bacteria collected.

**Response:** We added further details on the microbial culture and isolation processes in the revised manuscript.

Page 5, Line 134 – 144: "From each plate, all phenotypically distinct colonies were picked up and smeared onto fresh media for isolation. Single colonies were picked and re-streaked from each plate at least three times to isolate individual strains (Timm et al., 2020). Multiple streaking isolations are essential to obtain stable and pure strains by gradually diluting the microbial population and allowing for the isolation of individual colonies (Yan et al., 2021; Dunbar et al., 2015).

Isolated strains were preserved using the cryopreservation method. Bacterial strains were cultured in tryptic soy broth (TSB) at 37°C with shaking at 200 rpm, while fungal strains were cultured at 28°C on SDA medium plates under a stationary condition. When the bacteria reached the logarithmic phase, 750 μL of the culture was mixed thoroughly with an equal volume of 50% glycerol, resulting in a final glycerol concentration of 25%, and then stored at –80°C. For fungal strains, once spores were

evenly distributed across the plate, they were eluted with 4 mL of sterile PBS buffer. A 750 μL aliquot of the spore suspension was then mixed with an equal volume of 50% glycerol and stored at –80°C with a final glycerol concentration of 25%."

Page 5, Line 174 – 178: "The metabolic products (2 mL) filtered by 0.22 μm pore membranes were acidified to pH 2 using high-pressure liquid chromatography (HPLC) grade hydrochloric acid (HCl). Dissolved organic matter (DOM) was extracted using a solid phase extraction (SPE) cartridge (200 mg, Oasis HLB, 6cc, Waters, U.S.) to remove salts (Chen et al., 2022; Han et al., 2022). After extraction, the cartridges were dried by flushing with high-purity $N_2$. Finally, 6 mL of HPLC-grade methanol (Sigma-Aldrich) was used to elute the extracted DOM. A 2 mL aliquot of the eluent was collected and stored at –20°C for further analysis."

R3 Comment 5: Line 158-160, the culture time of bacteria is 7 days, and fungi is 15 days. Is there any reference for the culture time? Or refer to the residence time of microorganisms in the atmosphere? In Line 241, whether the results of different times are comparable compared to the results of culture for 30 days?

**Response:** Thank you for your comments. The choice of culture time was mainly based on the literatures on the viability and metabolic activities (e.g., organic biotransformation) of airborne bacteria and fungi, as well as their residence time in the atmosphere. Most experimental designs studying bacterial metabolites typically use an incubation time of 3 to 10 days. For example, Matulová et al. (2014) investigated the metabolic ability of *Bacillus* sp. 3B6, culturing the bacterial strain for 5 – 8 days to degrade different types of sugars and produce extracellular polymeric substances (EPSs). Similarly, Na et al. (2023) selected a 7-day incubation time to study the viability of atmospheric bacteria.

In addition, the kinetic study in microbial degradation of atmospheric organic compounds showed that about 15 days of incubation were required for the complete conversion of dicarboxylic acids (DCA) by fungi (Ariya et al., 2002). Some other studies on fungal metabolites used 10 – 20 days for fungal liquid fermentation (Woo et

al., 2014; Adelusi et al., 2022).

On the other hand, evidences suggest that atmospheric bacteria can reside for 2–15 days, enabling potential intercontinental transport of airborne bacteria (Lappan et al., 2024; Šantl-Temkiv et al., 2022). Laboratory simulations have further illustrated that both airborne bacteria and fungi maintain biological effectiveness for about 20 days, regardless of air pollution (Xu et al., 2021).

Considering these findings, we selected an experimental protocol of 7 days for bacterial incubation and 15 days for fungal incubation. Although incubation times vary across studies, microbial metabolic activities are continuous and dynamic processes. During these incubation periods, bioavailable organic molecules can undergo sufficient transformation, allowing the comparisons between different results meaningful.

Refrences:

Adelusi, O. A., Gbashi, S., Adebiyi, J. A., Makhuvele, R., Adebo, O. A., Aasa, A. O., Targuma, S., Kah, G., and Njobeh, P. B.: Variability in metabolites produced by *Talaromyces pinophilus* SPJ22 cultured on different substrates, Fungal Biology and Biotechnology, 9, https://doi.org/10.1186/s40694-022-00145-8, 2022.

Ariya, P. A., Nepotchatykh, O., Ignatova, O., and Amyot, M.: Microbiological degradation of atmospheric organic compounds, Geophysical Research Letters, 29, 2077, https://doi.org/10.1029/2002gl015637, 2002.

Lappan, R., Thakar, J., Molares Moncayo, L., Besser, A., Bradley, J. A., Goordial, J., Trembath-Reichert, E., and Greening, C.: The atmosphere: a transport medium or an active microbial ecosystem?, The ISME Journal, 18, https://doi.org/10.1093/ismejo/wrae092, 2024.

Matulová, M. r., Husárová, S., Capek, P., Sancelme, M., and Delort, A.-M.: Biotransformation of Various Saccharides and Production of Exopolymeric Substances by Cloud-Borne *Bacillus* sp. 3B6, Environmental Science & Technology, 48, 14238-14247, https://doi.org/10.1021/es501350s, 2014.

Na, H., Qi, J., Zhen, Y., Yao, X., and Gao, H.: Asian dust-transported bacteria survive in seawater and alter the community structures of coastal bacterioplankton in the Yellow Sea, Global and Planetary Change, 224, 104115, https://doi.org/10.1016/j.gloplacha.2023.104115, 2023.

Šantl-Temkiv, T., Amato, P., Casamayor, E. O., Lee, P. K. H., and Pointing, S. B.: Microbial ecology

of the atmosphere, FEMS Microbiology Reviews, 46, 1-18, https://doi.org/10.1093/femsre/fuac009, 2022.

Woo, P. C. Y., Lam, C.-W., Tam, E. W. T., Lee, K.-C., Yung, K. K. Y., Leung, C. K. F., Sze, K.-H., Lau, S. K. P., and Yuen, K.-Y.: The biosynthetic pathway for a thousand-year-old natural food colorant and citrinin in Penicillium marneffei, Scientific Reports, 4, https://doi.org/10.1038/srep06728, 2014.

Xu, C., Chen, H., Liu, Z., Sui, G., Li, D., Kan, H., Zhao, Z., Hu, W., and Chen, J.: The decay of airborne bacteria and fungi in a constant temperature and humidity test chamber, Environment International, 157, 106816, https://doi.org/10.1016/j.envint.2021.106816, 2021.

R3 Comment 6: Line 198, here discussed the genus of *Pantoea*, is *Pantoea* also a dominant bacterium? Another predominant strain, *Streptomyces*, was not further explained.

**Response:** Bacteria of the genus *Pantoea* were successfully isolated and characterized in our culture experiments, demonstrating the airborne bacterial cells were intact, viable, and culturable. In our ongoing studies, we analyzed the atmospheric bacterial community during the winter season in urban Tianjin through metagenomic sequencing. The results showed that *Pantoea* was the major bacterial taxa among the atmospheric bacteria (Fig. R2). Moreover, *Pantoea* is a well known genera containing potential ice-nucleating active bacterial species (Akila et al., 2018; Koda et al., 2000; Muryoi et al., 2003). Thereforewe discussed the genus *Pantoea* in this paper. Additionally, we added a discussion of the dominant bacterium, *Streptomyces*, highlighted in blue in the revised manuscript.

Page13, Line 403 – 407: "At the genus level, culturable atmospheric bacteria isolated from the aerosol samples mainly included *Bacillus*, *Pseudomonas*, *Streptomyces*, and *Pantoea* (Fig. 1). These bacterial taxa were predominant across urban, rural and forest aerosols, displaying strong environmental adaptability (Nunez et al., 2021; Gusareva et al., 2019; Souza et al., 2021). Their widespread distribution suggests remarkable resilience to harsh environmental stressors, enabling them to survive and maintain culturability (Hu et al., 2018; Hu et al., 2020)."

Page 14, Line 423 – 427: "*Streptomyces* is a dominant genus in aerosols from various megacities and background areas, especially in winter (Chen et al., 2021a; Petroselli et al., 2021; Li et al., 2019). Furthermore, *Streptomyces* has been widely reported in aerosols from European deserts and urban sewage treatment plants (Núñez et al., 2024; Zhan et al., 2024). *Streptomyces* is a drought-tolerant bacterial genus with spore-forming ability (Taketani et al., 2016). Their spores can enter a stable and quiescent state under environmental stress, enabling survival in diverse natural conditions (Naylor et al., 2017)."

[Figure]

**Figure R2** Relative abundance of bacterial composition during winter in urban Tianjin (TJ). (a) Bacterial composition at the genus level showing the top 50 genera; (b) Bacterial composition at the species level showing the top 30 species. WC, WM, and WP mean a sample collected on clean, moderate polluted, polluted winter days, respectively (unpublished data).


R3 Comment 7: Line 279, the diversity of exometabolomic products in bacteria and fungi is not analogous, hence a comparison between the two is not advisable. However, it is permissible to separately compare bacterial diversity across different samples and fungal diversity across different samples.

**Response:** We agree with your comment. We removed the comparison between the diversity of bacterial and fungal exometabolites, and added comparisons of exometabolite diversity among different bacterial and fungal species. Figure S4 was also modified accordingly.

Page 9, Line 268 – 274: "For bacteria, the molecular diversity of exometabolites varied considerably among species. The molecular Shannon diversity index of Gram-positive bacteria was 5.64, slightly lower than that of Gram-negative bacteria (6.19). Among the six bacterial strains, *Pantoea vagans*, *Streptomyces pratensis*, and *Stenotrophomonas* sp. showed higher molecular diversity in their exometabolites (Fig. S4a). For fungi, *Rhodotorula mucilaginosa* and *Aspergillus* species demonstrated higher molecular diversity of exometabolites (Fig. S4b). The two *Penicillium* species had similar molecular diversity, but the Chao1 index varied considerably. These results indicate the potential metabolic diversity of atmospheric bacteria and fungi."

R3 Comment 8: Line325, it is advised to omit the comparison between bacteria and fungi. section 3.4 "**Metabolic processes of typical isolated bacterial and fungal strains**", the discussion of metabolic processes in bacterial and fungal in 3.4.1 and 3.4.2 is separate. In fact, in the atmosphere, the metabolism of bacteria and fungi is closely related. The authors can add a discussion of correlations between bacterial and fungal metabolic processes, thus the influence of the metabolic process of bacteria and fungi

on atmospheric environment and biogeochemical cycles can be further clarified.

**Response:** Thank you for your comments. In this exploratory study, we selected 12 representative microorganisms from an urban atmospheric environment, including 6 bacterial and 6 fungal strains. Given the relatively small sample size, the correlation results may be biased by this limitation when assessing metabolite product correlations. Therefore, we analyzed and discussed the correlation between bacterial and fungal metabolic processes, and we explain but prefer not show the detail in the manuscript. Page 13, Line 397 – 400: "The metabolic processes of bacteria and fungi in the surface Earth system are closely linked, which is overlooked in this study, and further elucidation of these connections will enhance our understanding of the impact of microorganisms on atmospheric environments and biogeochemical processes." was added.

The specific results are as follows:

There were 200,716 correlations between bacterial and fungal metabolites with a correlation coefficient $|r| \geq 0.8$ and $p < 0.01$. The highest proportion of correlations (66.2%) were among bacteria-unique products, followed by those among bacteria-unique products and shared products of bacteria and fungi (20.8%) (Fig. R3a). This finding suggests that bacterial metabolites exhibited a high diversity and strong interconnections among metabolites from different bacteria. Within bacteria-unique products, only positive correlations existed, and the same for fungi-unique products. While, there were only negative correlations among bacteria-unique and fungi-unique products (Fig. R3b).

The percentage of correlations among shared products of bacteria and fungi was lower compared to the correlations among bacteria and fungi shared products and bacteria-unique products, both of which were dominated by positive correlations (Fig. R3b). Bacterial metabolites are essential in the interconnections of dissolved organic matter. The correlation network diagram shows that the bacteria and fungi shared products connected bacteria-unique and fungi-unique products, with most correlations being positive (Fig. R3c). Negative correlations were concentrated in a few shared

molecules, such as $C_{13}H_{10}O_6$, $C_{14}H_{12}O_6$, and $C_{16}H_{16}O_6$ (Fig. R3c). These shared products may serve as intermediaries between bacterial and fungal metabolites.

While many studies have explored the link between bacterial communities and dissolved organic matter, the role of fungi in organic matter transformation has often been overlooked. The results of this study offer valuable to support research on microbial contributions to the molecular composition of organic matter and interactions between organic molecules in nature ecosystems.

[Figure]

**Figure R3** Correlation between bacterial and fungal metabolites. Molecules that occur only in bacterial metabolites are labled as B_U, molecules that occur only in fungal metabolites are labled as F_U, and molecules that occur in both bacterial and fungal metabolites are labled as BF. (a) The pie charts characterize the overall distribution of results for metabolite correlations (Spearman's correlation coefficient $|r| \geq 0.8$, $p < 0.01$), for example, B_U vs F_U means the subgroup that shows the total number fraction of significant correlations between one bacterial-unique product and one fungal-unique

product. (b) The bar charts show the percentage of positive and negative correlations in different groups. (c) The network diagram shows the link between products shared by bacteria and fungi (BF) and products unique to bacteria (B_U) or fungi (F_U).

[revised manuscript text omitted]
 organic compounds in the initial culture broths. (a-b) The mass spectrum and elemental composition of TSB used for bacterial growth (a), and the formula numbers of different categories of organic matter for TSB (b). (c-d) The mass spectrum and elemental composition of SDB used for fungal growth (c), and the formula numbers of different categories of organic matter for SDB (d).

[Figure]

**Figure S4.** The molecular diversity of bacterial (a) and fungal (b) exometabolites.

[Figure]

**Figure S5.** The formula categories of bacterial exometabolites and differences in the molecular composition of shared and unique exometabolites among bacterial strains. (a) Stacked bars illustrate formula categories based on H/C and O/C ratios in bacterial exometabolites, and pie charts illustrate the elemental compositions of high abundance categories. (b) van Krevelen diagrams illustrate the shared molecules. Radar maps show the distribution of oxygen number of unique CHO compounds in exometabolites from *Pantoea vagans* (c) and the distribution of oxygen number of unique CHONS compounds in exometabolites from *Stenotrophomonas* sp. (d) and *Pseudomonas baetica* (e).

[Figure]

**Figure S6.** The formula categories of fungal exometabolites (a) and the distribution of oxygen number for unique CHONS compounds in exometabolites from *Rhodotorula mucilaginosa* (b). Stacked bars illustrate formula categories based on H/C and O/C ratios in fungal exometabolites, and pie charts illustrate the elemental compositions of high abundance categories.

[Figure]

**Figure S7.** Changes in the supernatant of the culture media for the genera *Penicillium* and *Talaromyces* after 14 days.

[Figure]

**Figure S8.** The differences in molecular transformations for bacterial and fungal species. (a-c) Venn diagrams display the numbers of shared and unique potential molecular transformations of bacteria and fungi (a), typical bacteria (b), and typical fungi (c). Each strain's total number of transformations is labeled below the species name. (d) Bar diagrams show the grouped distribution of the transformation numbers and the relative abundance of bacteria and fungi. (e) Bar diagram illustrates the distribution of transformation types with significant differences (Fold Change (FC) ≥ 2 or ≤ 0.5) between bacteria and fungi. Log2 (FC) ≥ 0 indicates that this transformation type is more significantly represented in bacteria.

[Figure]

**Figure S9.** The KEGG metabolic pathways of typical bacterial (a) and fungal (b) strains enriched and analyzed based on their exometabolites, including both primary and secondary pathways.

55  **Table S1** Compositions of the media used for isolation of culturable bacteria and fungi in the urban atmosphere.

| Component | Tryptic soy agar (TSA) | Sabouraud dextrose agar (SDA) |
|---|---|---|
| Tryptone | 15 g | - |
| Soy peptone | 5 g | - |
| NaCl | 5 g | - |
| Peptone | - | 10 g |
| Glucose | - | 40 g |
| Agar | 15 g | 20 g |
| Double distilled water | 1000 mL | 1000 mL |

**Table S2** Identities of cultivable bacteria derived from aerosol samples.

| Isolate ID | Isolated strains | BLAST Identity | % Identity | Phylum | Class | Order | Family |
|---|---|---|---|---|---|---|---|
| B1 | *Bacillus* sp. TJB1 | *Bacillus* sp. | 99.93% | Bacillota | Bacilli | Bacillales | Bacillaceae |
| B2 | *Pantoea vagans* TJB2 | *Pantoea vagans* | 100.00% | Pseudomonadota | Gammaproteobacteria | Enterobacterales | Erwiniaceae |
| B3 | *Streptomyces thermoviolaceus* TJB3 | *Streptomyces thermoviolaceus* | 100.00% | Actinomycetota | Actinobacteria | Streptomycetales | Streptomycetaceae |
| B4 | *Bacillus subtilis* TJB4 | *Bacillus subtilis* | 99.17% | Bacillota | Bacilli | Bacillales | Bacillaceae |
| B5 | *Streptomyces thermoviolaceus* TJB5 | *Streptomyces thermoviolaceus* | 99.93% | Actinomycetota | Actinobacteria | Streptomycetales | Streptomycetaceae |
| B6 | *Bacillus* sp. TJB6 | *Bacillus* sp. Y1(2012) | 99.93% | Bacillota | Bacilli | Bacillales | Bacillaceae |
| B7 | *Bacillus subtilis* TJB7 | *Bacillus subtilis* | 100.00% | Bacillota | Bacilli | Bacillales | Bacillaceae |
| B8 | *Erwinia* sp. TJB8 | *Erwinia* sp. | 100.00% | Pseudomonadota | Gammaproteobacteria | Enterobacterales | Erwiniaceae |
| B9 | *Erwinia* sp. TJB9 | *Erwinia* sp. | 100.00% | Pseudomonadota | Gammaproteobacteria | Enterobacterales | Erwiniaceae |
| B10 | *Streptomyces pratensis* TJB10 | *Streptomyces pratensis* | 99.12% | Actinomycetota | Actinobacteria | Streptomycetales | Streptomycetaceae |
| B11 | *Streptomyces* sp. TJB11 | *Streptomyces* sp. SYP-A7193 | 99.93% | Actinomycetota | Actinobacteria | Streptomycetales | Streptomycetaceae |
| B12 | *Bacillus* sp. TJB12 | *Bacillus* sp. 210_62 | 99.72% | Bacillota | Bacilli | Bacillales | Bacillaceae |
| B13 | *Pseudomonas* sp. TJB13 | *Pseudomonas* sp. | 99.93% | Pseudomonadota | Gammaproteobacteria | Pseudomonadales | Pseudomonadaceae |
| B14 | *Pantoea vagans* TJB14 | *Pantoea vagans* | 99.32% | Pseudomonadota | Gammaproteobacteria | Enterobacterales | Erwiniaceae |
| B15 | *Pseudomonas* sp. TJB15 | *Pseudomonas* sp. B14-6 | 100.00% | Pseudomonadota | Gammaproteobacteria | Pseudomonadales | Pseudomonadaceae |
| B16 | *Stenotrophomonas* sp. TJB16 | *Stenotrophomonas* sp. | 99.79% | Pseudomonadota | Gammaproteobacteria | Xanthomonadales | Xanthomonadaceae |
| B17 | *Pseudomonas fluorescens* TJB17 | *Pseudomonas fluorescens* | 100.00% | Pseudomonadota | Gammaproteobacteria | Pseudomonadales | Pseudomonadaceae |
| B18 | *Bacillus toyonensis* TJB18 | *Bacillus toyonensis* | 100.00% | Bacillota | Bacilli | Bacillales | Bacillaceae |
| B19 | *Pseudomonas* sp. TJB19 | *Pseudomonas* sp. | 99.71% | Pseudomonadota | Gammaproteobacteria | Pseudomonadales | Pseudomonadaceae |
| B20 | *Bacillus halotolerans* TJB20 | *Bacillus halotolerans* | 100.00% | Bacillota | Bacilli | Bacillales | Bacillaceae |
| B21 | *Bacillus proteolyticus* TJB21 | *Bacillus proteolyticus* | 100.00% | Bacillota | Bacilli | Bacillales | Bacillaceae |
| B22 | *Bacillus safensis* TJB22 | *Bacillus safensis* | 100.00% | Bacillota | Bacilli | Bacillales | Bacillaceae |
| B23 | *Pseudomonas baetica* TJB23 | *Pseudomonas baetica* | 100.00% | Pseudomonadota | Gammaproteobacteria | Pseudomonadales | Pseudomonadaceae |
| B24 | *Pseudomonas* sp. TJB24 | *Pseudomonas* sp. | 99.85% | Pseudomonadota | Gammaproteobacteria | Pseudomonadales | Pseudomonadaceae |

**Table S3** Identities of cultivable fungi derived from aerosol samples.

| Isolate ID | Isolated strains | BLAST Identity | % Identity | Phylum | Class | Order | Family |
|---|---|---|---|---|---|---|---|
| F1 | *Trametes elegans* TJF1 | *Trametes elegans* | 100.00% | Basidiomycota | Agaricomycetes | Polyporales | Polyporaceae |
| F2 | *Talaromyces* sp. TJF2 | *Talaromyces* sp. | 100.00% | Ascomycota | Eurotiomycetes | Eurotiales | Trichocomaceae |
| F3 | *Aspergillus niger* TJF3 | *Aspergillus niger* | 100.00% | Ascomycota | Eurotiomycetes | Eurotiales | Aspergillaceae |
| F4 | *Aspergillus* sp. TJF4 | *Aspergillus* sp. | 100.00% | Ascomycota | Eurotiomycetes | Eurotiales | Aspergillaceae |
| F5 | *Aspergillus* sp. TJF5 | *Aspergillus* sp. | 100.00% | Ascomycota | Eurotiomycetes | Eurotiales | Aspergillaceae |
| F6 | *Aspergillus nidulans* TJF6 | *Aspergillus nidulans* | 100.00% | Ascomycota | Eurotiomycetes | Eurotiales | Aspergillaceae |
| F7 | *Penicillium oxalicum* TJF7 | *Penicillium oxalicum* | 100.00% | Ascomycota | Eurotiomycetes | Eurotiales | Aspergillaceae |
| F8 | *Sarocladium terricola* TJF8 | *Sarocladium terricola* | 100.00% | Ascomycota | Sordariomycetes | Hypocreales | Sarocladiaceae |
| F9 | *Penicillium sumatraense* TJF9 | *Penicillium sumatraense* | 100.00% | Ascomycota | Eurotiomycetes | Eurotiales | Aspergillaceae |
| F10 | *Penicillium aurantiogriseum* TJF10 | *Penicillium aurantiogriseum* | 100.00% | Ascomycota | Eurotiomycetes | Eurotiales | Aspergillaceae |
| F11 | *Aspergillus* sp. TJF11 | *Aspergillus* sp. | 100.00% | Ascomycota | Eurotiomycetes | Eurotiales | Aspergillaceae |
| F12 | *Aspergillus carneus* TJF12 | *Aspergillus carneus* | 100.00% | Ascomycota | Eurotiomycetes | Eurotiales | Aspergillaceae |
| F13 | *Cladosporium parahalotolerans* TJF13 | *Cladosporium parahalotolerans* | 100.00% | Ascomycota | Dothideomycetes | Cladosporiales | Cladosporiaceae |
| F14 | *Rhodotorula mucilaginosa* TJF14 | *Rhodotorula mucilaginosa* | 100.00% | Basidiomycota | Microbotryomycetes | Sporidiobolales | Sporidiobolaceae |
| F15 | *Penicillium cinnamopurpureum* TJF15 | *Penicillium cinnamopurpureum* | 100.00% | Ascomycota | Eurotiomycetes | Eurotiales | Aspergillaceae |
| F16 | *Cladosporium* sp. TJF16 | *Cladosporium* sp. | 99.42% | Ascomycota | Dothideomycetes | Cladosporiales | Cladosporiaceae |

**Table S4** The bacterial and fungal isolates used for exometabolomic studies.

| Kingdom | Isolate ID | Isolated strains | Species | Category |
|---|---|---|---|---|
| Bacteria | B10 | *Streptomyces pratensis* TJB10 | *Streptomyces pratensis* | Gram-positive bacteria |
| | B4 | *Bacillus subtilis* TJB4 | *Bacillus subtilis* | Gram-positive bacteria |
| | B18 | *Bacillus toyonensis* TJB18 | *Bacillus toyonensis* | Gram-positive bacteria |
| | B2 | *Pantoea vagans* TJB2 | *Pantoea vagans* | Gram-negative bacteria |
| | B16 | *Stenotrophomonas* sp. TJB16 | *Stenotrophomonas* sp. | Gram-negative bacteria |
| | B23 | *Pseudomonas baetica* TJB23 | *Pseudomonas baetica* | Gram-negative bacteria |
| Fungi | F14 | *Rhodotorula mucilaginosa* TJF14 | *Rhodotorula mucilaginosa* | yeast |
| | F3 | *Aspergillus niger* TJF3 | *Aspergillus niger* | mold |
| | F5 | *Aspergillus* sp. TJF5 | *Aspergillus* sp. | mold |
| | F10 | *Penicillium aurantiogriseum* TJF10 | *Penicillium aurantiogriseum* | mold |
| | F7 | *Penicillium oxalicum* TJF7 | *Penicillium oxalicum* | mold |
| | F2 | *Talaromyces* sp. TJF2 | *Talaromyces* sp. | mold |

Note: The species names instead of the strain names are used in the figures and tables to facilitate the description of the different isolates.

**Table S5** Changes in the number of formulas after incubation for bacteria and fungi.

| Media / Species | Culture nutrients | Consumed | | | Resistant | | | Produced | | | Total |
|---|---|---|---|---|---|---|---|---|---|---|---|
| | | Number | H/C | O/C | Number | H/C | O/C | Number | H/C | O/C | Number |
| **Before incubation** | | | | | | | | | | | |
| TSB | - | - | | | - | | | - | | | 3416 |
| SDB | - | - | | | - | | | - | | | 3920 |
| **After incubation** | | | | | | | | | | | |
| *Streptomyces pratensis* | TSB | 689 | 1.60 | 0.31 | 2727 | 1.46 | 0.39 | 2158 | 1.32 | 0.46 | 4885 |
| *Bacillus subtilis* | TSB | 477 | 1.56 | 0.30 | 2939 | 1.47 | 0.39 | 1479 | 1.43 | 0.44 | 4418 |
| *Bacillus toyonensis* | TSB | 708 | 1.58 | 0.32 | 2708 | 1.46 | 0.39 | 1596 | 1.44 | 0.44 | 4304 |
| *Pantoea vagans* | TSB | 413 | 1.58 | 0.30 | 3003 | 1.47 | 0.39 | 1761 | 1.32 | 0.45 | 4764 |
| *Stenotrophomonas* sp. | TSB | 1198 | 1.57 | 0.36 | 2218 | 1.44 | 0.39 | 2868 | 1.42 | 0.46 | 5086 |
| *Pseudomonas baetica* | TSB | 1122 | 1.53 | 0.34 | 2294 | 1.47 | 0.39 | 651 | 1.65 | 0.53 | 2945 |
| Mean ± SE | | 768±133 | 1.57±0.01 | 0.32±0.01 | 2648±133 | 1.46±0.00 | 0.39±0.00 | 1752±301 | 1.43±0.05 | 0.46±0.01 | 4400±314 |
| **After incubation** | | | | | | | | | | | |
| *Rhodotorula mucilaginosa* | SDB | 811 | 1.56 | 0.29 | 3109 | 1.45 | 0.40 | 971 | 1.39 | 0.48 | 4080 |
| *Aspergillus niger* | SDB | 1689 | 1.43 | 0.36 | 2231 | 1.50 | 0.39 | 328 | 1.53 | 0.40 | 2559 |
| *Aspergillus* sp. | SDB | 2049 | 1.50 | 0.34 | 1871 | 1.43 | 0.43 | 503 | 1.32 | 0.49 | 2374 |
| *Penicillium aurantiogriseum* | SDB | 1262 | 1.55 | 0.33 | 2658 | 1.43 | 0.40 | 1501 | 1.26 | 0.44 | 4159 |
| *Penicillium oxalicum* | SDB | 1362 | 1.52 | 0.35 | 2558 | 1.44 | 0.40 | 460 | 1.33 | 0.39 | 3018 |
| *Talaromyces* sp. | SDB | 2531 | 1.48 | 0.36 | 1389 | 1.45 | 0.41 | 246 | 1.06 | 0.44 | 1635 |
| Mean ± SE | | 1617±250 | 1.51±0.02 | 0.34±0.01 | 2303±250 | 1.45±0.01 | 0.41±0.01 | 668±196 | 1.32±0.06 | 0.44±0.02 | 2971±406 |

**Table S6** The molecular weight distribution of the typical bacterial and fungal exometabolites.

| Species | Low Molecular Weight (LMW) (150 – 300 Da) | Medium Molecular Weight (MMW) (300 – 500 Da) | High Molecular Weight (HMW) (500 – 800 Da) | Total |
|---|---|---|---|---|
| **Bacteria** | | | | |
| *Streptomyces pratensis* | 665 30.8% | 1034 47.9% | 459 21.3% | 2158 100% |
| *Bacillus subtilis* | 460 31.1% | 731 49.4% | 288 19.5% | 1479 100% |
| *Bacillus toyonensis* | 540 33.8% | 774 48.5% | 282 17.7% | 1596 100% |
| *Pantoea vagans* | 617 35.1% | 837 47.5% | 307 17.4% | 1761 100% |
| *Stenotrophomonas* sp. | 721 25.1% | 1244 43.4% | 903 31.5% | 2868 100% |
| *Pseudomonas baetica* | 123 18.9% | 252 38.7% | 276 42.4% | 651 100% |
| **Fungi** | | | | |
| *Rhodotorula mucilaginosa* | 318 32.7% | 518 53.4% | 135 13.9% | 971 100% |
| *Aspergillus niger* | 55 16.8% | 155 47.3% | 118 35.9% | 328 100% |
| *Aspergillus* sp. | 78 15.5% | 299 59.4% | 126 25.1% | 503 100% |
| *Penicillium aurantiogriseum* | 187 12.5% | 929 61.9% | 385 26.6% | 1501 100% |
| *Penicillium oxalicum* | 100 21.7% | 222 48.3% | 138 30.0% | 460 100% |
| *Talaromyces* sp. | 54 21.9% | 164 66.7% | 28 11.4% | 246 100% |

**Table S7** The elemental compositions of the typical bacterial and fungal exometabolites.

| Species | CHO | CHON | CHOS | CHONS | CHONSP | Total |
|---|---|---|---|---|---|---|
| **Bacteria** | | | | | | |
| *Streptomyces pratensis* | 137
6.4% | 1552
71.9% | 43
2.0% | 426
19.7% | 0
0.0% | 2158
100% |
| *Bacillus subtilis* | 150
10.1% | 1022
69.1% | 37
2.5% | 270
18.3% | 0
0.0% | 1479
100% |
| *Bacillus toyonensis* | 134
8.4% | 1070
67.0% | 52
3.3% | 340
21.3% | 0
0.0% | 1596
100% |
| *Pantoea vagans* | 262
14.9% | 1131
64.2% | 45
2.6% | 318
18.1% | 5
0.2% | 1761
100% |
| *Stenotrophomonas* sp. | 140
4.9% | 1494
52.1% | 69
2.4% | 1165
40.6% | 0
0.0% | 2868
100% |
| *Pseudomonas baetica* | 25
3.8% | 405
62.2% | 5
0.8% | 216
33.2% | 0
0.0% | 651
100% |
| **Fungi** | | | | | | |
| *Rhodotorula mucilaginosa* | 203
20.9% | 525
54.1% | 58
6.0% | 185
19.0% | 0
0.0% | 971
100% |
| *Aspergillus niger* | 80
24.4% | 233
71.0% | 15
4.6% | 0
0.0% | 0
0.0% | 328
100% |
| *Aspergillus* sp. | 120
23.9% | 377
74.9% | 5
1.0% | 1
0.2% | 0
0.0% | 503
100% |
| *Penicillium aurantiogriseum* | 402
26.8% | 1017
67.7% | 71
4.7% | 10
0.7% | 1
0.1% | 1501
100% |
| *Penicillium oxalicum* | 94
20.4% | 366
79.6% | 0
0.0% | 0
0.0% | 0
0.0% | 460
100% |
| *Talaromyces* sp. | 182
74.0% | 64
26.0% | 0
0.0% | 0
0.0% | 0
0.0% | 246
100% |

**Table S8** The formulas of possible pigments in exometabolites from *Penicillium* and *Talaromyces*.

| Genus | Species | Pigment | Formula |
|---|---|---|---|
| *Penicillium* | *Penicillium oxalicum* | Carviolin | $C_{16}H_{12}O_6$ |
| | | Phoenicin | $C_{14}H_{10}O_6$ |
| | *Penicillium aurantiogriseum* | Naphthalic anhydride | $C_{21}H_{22}O_9$ |
| | | Carviolin | $C_{16}H_{12}O_6$ |
| | | Phoenicin | $C_{14}H_{10}O_6$ |
| | | Atrovenetin | $C_{19}H_{18}O_6$ |
| | | (10Z)-12-carboxymonascorubrin (PP-O) | $C_{23}H_{24}O_7$ |
| | | N-glutarylrubropunctamine | $C_{26}H_{29}NO_8$ |
| | | Purpurquinone-A | $C_{21}H_{20}O_9$ |
| | | Northerqueinone | $C_{19}H_{18}O_7$ |
| | | Xanthomonasin A | $C_{21}H_{24}O_7$ |
| *Talaromyces* | *Talaromyces* sp. | Trihydroxymethyl-antraquinone | $C_{15}H_{10}O_5$ |
| | | Phoenicin | $C_{14}H_{10}O_6$ |
| | | Carviolin | $C_{16}H_{12}O_6$ |
| | | Naphthalic anhydride | $C_{21}H_{22}O_9$ |
| | | Purpurquinone-A | $C_{21}H_{20}O_9$ |
| | | Northerqueinone | $C_{19}H_{18}O_7$ |
| | | Mitorubrinol | $C_{21}H_{18}O_8$ |
| | | Atrovenetin | $C_{19}H_{18}O_6$ |

---

## Author Response (AR2)

Dear Editor and Reviewer,

We thank for your careful reading and constructive comments, which have allowed us to make further enhancements and significantly raised the quality of the manuscript. Our responses to the comments from the referee are provided below. The reviewer's comments are shown in blue, while our responses are in black. All page and line numbers mentioned in the response refer to the revised version of the manuscript. Revisions are marked in blue in the revised manuscript.

Sincerely,

HU Wei (on behalf of co-authors)

**Response to Referee 2 and Editor:**

**General comments:**

Comment 1: This paper has been revised in dependence on the reviewers' comments. I think this manuscript can be accepted after same revisions relating to the description about the biases of microbial isolation.

**Response:** Thank you very much for your valuable comments. We have double-checked and revised the manuscript with respect to the names and abbreviations of the isolated strains.

**Some major comments:**

Comment 2: The compounds of few microbial isolates from aerosol samples are discussed mainly in this study to suggested the biological compounds transported in atmosphere in Asia. In contrast, almost airborne microorganisms are known to be unculturable but viable microorganisms (In general, >90% of microorganisms cannot be cultured but they are living). The authors should indicate the occupation rate of the isolate species in atmospheric microorganisms with the calculations using the previous metadata bases, which are obtained at previous investigation at China continent.

**Response:** Thank you for your valuable comments. We appreciate the important point regarding the predominance of unculturable yet viable microorganisms in the atmosphere. In our study, however, the bacterial and fungal isolates we selected are representative of high-abundance and highly active species within the atmospheric microbiome, which are believed to represent a major fraction of biological compounds transported through the atmosphere.

While we performed extensive metagenomic analysis, the samples used for metagenomics did not exactly correspond to those from which the isolates were obtained. Nevertheless, based on our metagenomic results (unpublished data), we have estimated the proportion of the genera of isolates analyzed in this study within the broader atmospheric microbiome. Specifically, we estimate that the genera of isolates selected in our study represent approximately 9.8% of the airborne microorganisms in terms of the transcript per kilobase per million mapped reads (TPM) in winter in Tianjin.

This estimation, based on metagenomic data, further underscores the relevance of the cultured isolates in representing the atmospheric microbial community. Please refer to our former response to your comment. We hope that this additional information clarifies the relative abundance of the isolated genera within the context of the overall atmospheric microbial community.

**Some minor comments:**

Comment 3: L18: What is CHON?

**Response:** According to the elemental composition of organic compounds, their molecular composition is classified into four categories: CHO, CHON, CHOS, and CHONS. This classification has been commonly used in mass spectrometry analyses. For example, CHON means that the organic compounds contain only four elements: carbon, hydrogen, oxygen, and nitrogen without any other elements. Please refer to Line 199-201 in the revised manuscript.

Comment 4: L74: This survey has been performed at Asian regions. The bioaerosols around Asian regions should be also mentioned here using relative researches.

The pioneer work performed at Asian continental areas: Tang et al. ACP, 18, 7131-7148, 2018, doi.org/10.5194/acp-18-7131-2018

Asian bioaerosol review: Huang et al., STOTEN, 912, 168818, 2024. 168818, https://doi.org/10.1016/j.scitotenv.2023.168818

**Response:** Thank you for your comments. We appreciate the reviewer's recommendation to include references to previous studies on bioaerosols in the Asian region. In response, we have incorporated the relevant studies you mentioned into our revised manuscript. Specifically, we have added the following citations:

Tang, K., Huang, Z., Huang, J., Maki, T., Zhang, S., Shimizu, A., Ma, X., Shi, J., Bi, J., Zhou, T., Wang, G., and Zhang, L.: Characterization of atmospheric bioaerosols along the transport pathway of Asian dust during the Dust-Bioaerosol 2016 Campaign, Atmospheric Chemistry and Physics, 18, 7131-7148, https://doi.org/10.5194/acp-18-7131-2018, 2018. This pioneering study provides valuable insights into bioaerosols in

Asian continental areas.

Huang, Z., Yu, X., Liu, Q., Maki, T., Alam, K., Wang, Y., Xue, F., Tang, S., Du, P., Dong, Q., Wang, D., and Huang, J.: Bioaerosols in the atmosphere: A comprehensive review on detection methods, concentration and influencing factors, Science of The Total Environment, 912, https://doi.org/10.1016/j.scitotenv.2023.168818, 2024. This comprehensive review summarizes the composition and distribution characteristics of bioaerosols, providing a richer background for our study and strengthening the context for understanding bioaerosol behavior in the atmosphere, particularly in Asian regions. These additions are now included in the relevant sections of our manuscript to provide a broader context for our study and highlight the significance of bioaerosols in the Asian atmospheric environment.

Page 2, Line 44 – 46: "Atmospheric bacteria and fungi can maintain metabolic activities due to specific growth characteristics, such as spore production capacity, ultraviolet resistance, drought resistance, or through extracellular secretions (Huang et al., 2024; Matulová et al., 2014; Bryan et al., 2019)."

Page 2, Line 48 – 50: "Airborne microorganisms (especially those in Asian dust) form aggregates with organic matter, which may serve as a nutrient source, promoting microbial survival and facilitating long distance transport (Tang et al., 2018; Huang et al., 2024)."

Comment 5: L235: After the second indications of taxon name, the genera has to be abbreviated in biological research field. For example, *P. baetica*. Please correct same irregular usages at the other parts.

**Response:** Thank you for pointing out this issue. We have carefully reviewed the manuscript and corrected all instances where the genus names were not properly abbreviated after the first mention. All taxonomic names have now been revised to conform to the standard practice in biological research, such as "*P. baetica*" for *Pseudomonas baetica*, "*B. subtilis*" for *Bacillus subtilis*. We believe these corrections improve the consistency and accuracy of the manuscript. We appreciate your attention

to the detail and hope these revisions meet your expectation.

Comment 6: L276: *Aspergillus* spp. or *Aspergillus* sp. Species name is needed for matching to the forward taxon name.

**Response:** Thanks. We have carefully reviewed the manuscript and have now ensured that species names are used correctly and consistently throughout. The taxon names have been updated to ensure full consistency and completeness, with the appropriate species names included where necessary.

Page 9, Line 273 – 274: "For fungi, *Rhodotorula mucilaginosa*, *Aspergillus niger*, and *Aspergillus* sp. demonstrated higher molecular diversity of exometabolites (Fig. S4b)."

Comment 7: L448-L450: In introduction, the authors also introduce the biochemical cycle in clouds. How is this topic in conclusion? Additionally, the impact on freshwater?

**Response:** Thank you for your valuable comment. In our study, we highlight the potential role of atmospheric microorganisms in biogeochemical cycles, particularly their involvement in organic matter transformation during both transport and deposition. Atmospheric microorganisms can remain active in the air, cloud water, fog, and rainwater, where they contribute to carbon metabolism. This underscores their importance in biogeochemical processes within the atmosphere.

Regarding the conclusions, we have indeed addressed the impact of atmospheric microbial deposition on the biogeochemical cycles of freshwater ecosystems. These processes are crucial for understanding the broader ecological implications of atmospheric microorganisms. We have also ensured that the conclusion provides a clearer and more comprehensive summary of our findings and their environmental relevance.